# Tumour heterogeneity and intercellular networks of nasopharyngeal carcinoma at single cell resolution

Yang Liu[1,11], Shuai He [1,2,3,11], Xi-Liang Wang[4,11], Wan Peng[1], Qiu-Yan Chen[1], Dong-Mei Chi[1,5], Jie-Rong Chen[1,6], Bo-Wei Han[1], Guo-Wang Lin[1,7], Yi-Qi Li[1], Qian-Yu Wang [1], Rou-Jun Peng[1], Pan-Pan Wei[1], Xiang Guo[1], Bo Li [8,9], Xiaojun Xia [1], Hai-Qiang Mai[1], Xue-Da Hu[4], Zemin Zhang [4,10✉], Yi-Xin Zeng [1✉] & Jin-Xin Bei [1,2,3✉]

The heterogeneous nature of tumour microenvironment (TME) underlying diverse treatment responses remains unclear in nasopharyngeal carcinoma (NPC). Here, we profile 176,447 cells from 10 NPC tumour-blood pairs, using single-cell transcriptome coupled with T cell receptor sequencing. Our analyses reveal 53 cell subtypes, including tumour-infiltrating CD8[+] T, regulatory T (Treg), and dendritic cells (DCs), as well as malignant cells with different Epstein-Barr virus infection status. Trajectory analyses reveal exhausted CD8[+] T and immune-suppressive TNFRSF4[+] Treg cells in tumours might derive from peripheral CX3CR1[+]CD8[+] T and naïve Treg cells, respectively. Moreover, we identify immune-regulatory and tolerogenic LAMP3[+] DCs. Noteworthily, we observe intensive inter-cell interactions among LAMP3[+] DCs, Treg, exhausted CD8[+] T, and malignant cells, suggesting potential cross-talks to foster an immune-suppressive niche for the TME. Collectively, our study uncovers the heterogeneity and interacting molecules of the TME in NPC at single-cell resolution, which provide insights into the mechanisms underlying NPC progression and the development of precise therapies for NPC.

[1] Sun Yat-sen University Cancer Center, State Key Laboratory of Oncology in South China, Collaborative Innovation Center for Cancer Medicine, Guangdong Key Laboratory of Nasopharyngeal Carcinoma Diagnosis and Therapy, Guangzhou 510060, People's Republic of China. [2] The Eighth Affiliated Hospital, Sun Yat-sen University, Shenzhen 518033, People's Republic of China. [3] Center for Precision Medicine, Sun Yat-sen University, Guangzhou 510080, People's Republic of China. [4] BIOPIC, Beijing Advanced Innovation Center for Genomics, School of Life Sciences, Peking University, Beijing 100871, People's Republic of China. [5] Department of Anesthesiology, Sun Yat-sen University Cancer Center, Guangzhou 510060, People's Republic of China. [6] Guangdong General Hospital, Guangdong Academy of Medical Sciences, Guangzhou 510080, People's Republic of China. [7] Department of Laboratory Medicine, Zhujiang Hospital, Southern Medical University, Guangzhou, Guangdong 510282, People's Republic of China. [8] Department of Biochemistry and Molecular Biology, Zhongshan School of Medicine, Sun Yat-sen University, Guangzhou 510080, People's Republic of China. [9] RNA Biomedical Institute, Sun Yat-sen Memorial Hospital, Sun Yat-sen University, Guangzhou 510120, People's Republic of China. [10] Peking-Tsinghua Center for Life Sciences, Academy for Advanced Interdisciplinary Studies, Peking University, Beijing 100871, People's Republic of China. [11] These authors contributed equally: Yang Liu, Shuai He, Xi-Liang Wang. ✉email: zemin@pku.edu.cn; zengyx@sysucc.org.cn; beijx@sysucc.org.cn

Nasopharyngeal carcinoma (NPC) is a distinct type of head and neck cancer, which has been closely linked with the infection of Epstein-Barr virus (EBV)[1]. NPC has a remarkable ethnic and geographic prevalence, where high incidence rate of 15–50 cases per 100,000 people was reported in Southern China and Southeast Asia, as compared to 0.4 per 100,000 in western populations[2]. In general, patients with NPC are diagnosed with advanced stages largely due to non-specific symptoms. Radiotherapy is the primary treatment modality for patients with NPC because of the radiosensitive nature of its tumour cells[2]. Survival outcomes of patients with NPC improve substantially, reaching a 5-year overall survival rate of 85.6%[3,4], mainly benefited from the evolution of radiotherapy techniques and the addition of platinum-based chemotherapy in patients with loco-regionally advanced disease. Nevertheless, more than 10% of patients develop recurrent and metastatic NPC[2], for whom recent studies showed an overall response rate of 11.7% and 25.9–34% to targeted therapy with the addition of an inhibitor for epidermal growth factor receptor[5] and immunotherapies using the immune checkpoint blockade[6,7], respectively. These variations in treatment responses and survival outcomes indicate the heterogeneous nature of NPC.

Individual genetic makeup and genomic instability foster genetic diversity of cancer cells that contribute to tumour heterogeneity. Genome sequencing analyses have revealed diverse profiles of somatic alterations in NPC tumours, with high mutational frequencies at *CYLD*, *NFKBIA*, *TP53*, and *CDKN2A/B*, as well as accumulated mutations in MHC class I genes and chromatin modification genes, which were associated with poor overall survival of the patients[8,9]. Apart from heterogeneous cancer cells, tumours exhibit another dimension of heterogeneity, which contain diverse normal cells creating the tumour microenvironment (TME) for the maintenance of cancer hallmarks[10]. Heterogeneous immune cells and stromal cells have been characterised using transcriptional profiling at single-cell resolution in several cancers, revealing that certain subtypes of immune cells and gene signatures in TME are important for tumour progression and sustained treatment responses[11,12]. Profound infiltration of lymphocytes has been observed in histological biopsies of NPC, amid other stromal cells and tumour cells of different morphology[13,14]. Moreover, high density of tumour infiltrating lymphocytes (TILs) was associated with favourable survival outcomes of patients with NPC[15,16]. However, the composition of diverse cell populations in the TME has not been well illustrated in NPC. Two recent studies demonstrated the presence of T cells with various functional states and different immune cells in NPC tumours, using single-cell transcriptome analysis[17,18].

In this work, we aim to provide a comprehensive global view of tumour heterogeneity of NPC, by analysing the single-cell transcriptional profiles of 176,447 cells from 10 treatment-naïve patients with NPC. With the combination of T cell receptor (TCR) repertoire information and individual tumour-blood sample pairs, we further characterise the clonality and migrations of T cells. In addition, we generate a potential cellular interaction network of cell populations in the TME of NPC.

## Results

**Landscape view of cell composition in tumour biopsy and PBMC in patients with NPC.** To shed light on the complexity of tumour microenvironment (TME) in NPC, we performed single-cell RNA sequencing in combination of TCR repertoire sequencing on viable cells derived from tumour biopsies and matched peripheral blood mononuclear cells (PBMC) for 10 patients with EBV-positive NPC prior to any anti-cancer treatment (Fig. 1a, Supplementary Fig. 1a, b, and Supplementary Table 1). On average, we obtained more than 380 million sequencing reads for each sample, with the median of sequencing saturation (covering the fraction of library complexity) at 90.75% (75.90%–94.50%; Supplementary Table 2). After strict quality control filters (low expression of representative genes and inferred doublets; see Methods; Supplementary Fig. 1c and Supplementary Table 3), a total of 176,447 cells were identified from the 10 patients (including 82,622 and 93,825 for tumours and PBMC, respectively; Supplementary Data 1). We obtained about 1500 genes and 4950 unique molecular identifiers (UMIs) on average for each cell, indicating sufficient coverage and representative of transcripts (Supplementary Fig. 1d and Supplementary Data 1).

Next, to define groups of cells with similar expression profiles, we performed unsupervised clustering analysis implemented in Seurat software[19]. The distribution of cell clusters for each patient matched well with that of other patients, suggesting that the potential variation of expression due to batch effect of sample processing was minimal (see Methods; Supplementary Fig. 1e). Each cluster was further identified as a specific cell subpopulation according to the expression of the most variable genes and the canonical markers, including CD4+ T cells (gene markers: *PTPRC*, *CD3D*, and *CD4*), CD8+ T cells (*PTPRC*, *CD3D*, and *CD8A*), myeloid cells (*CD14*, *ITGAX* for CD11C), malignant cells (*EPCAM* and *KRT5*), B cells (*CD19* and *MS4A1*), and NK cells (*FCGR3A* and *NCAM1*; Fig. 1b). Besides, we detected 56 fibroblasts and seven endothelial cells with sparse distribution among TME cells (Supplementary Fig. 1f), which might reflect their intrinsic nature in NPC or their low representation due to technical limitations. All these cell types were widespread in tumour samples, indicating the heterogeneous cell composition of TME in NPC (Fig. 1b), consistent with a recent single-cell transcriptome study of NPC[17]. We observed that the proportions of CD8+ T and B cells were increased with 1.34 and 2.33 times on average, respectively, while the NK cells were decreased in the tumours compared to the PBMC, suggesting two distinct immune landscapes between tumour and peripheral blood (Fig. 1c). Moreover, we compared cell compositions between NPC and other types of cancers with single-cell data publicly available, including non-small-cell lung cancer (NSCLC), colorectal cancer (CRC), and pancreatic ductal adenocarcinoma (PDAC). We observed the common occurrence of infiltrating immune cells amid individual heterogeneity of cell composition in all types of cancers (Supplementary Fig. 1g). Notably, we observed a significantly higher proportion of T cells in NPC compared to any other cancers (Supplementary Fig. 1h), which is consistent with a previous finding that the tumour infiltrating leucocytes was the main characteristic of NPC stroma[20].

**Heterogeneity of T cells and the diversity of TCR repertoire.** Considering the abundance of T and NK cells in NPC tumour samples and their anti-tumour capabilities, we explored the intrinsic structure and potential functional subtypes of the overall T and NK cell populations. We grouped all 141,875 T and NK cells into 32 subgroups using clustering analysis, of which the majority were CD4+ and CD8+ T cells (Fig. 2a, Supplementary Fig. 2a, Supplementary Data 1 and 2). To identify any gene with a specific expression on a cell type, we performed differential gene expression (DEG) analysis of T cell clusters. We observed that CD4+ and CD8+ T cell clusters in tumour samples had widespread overexpression of exhaustion markers (*LAG3*, *TIGIT*, *PDCD1*, *HAVCR2*, and *CTLA4*) and effector molecules (*GZMB*, *GZMK*, *INFG*, *NKG7*, *GNLY*, and *IL2*; Supplementary Fig. 2b), with remarkably high expression of the proliferative signatures for CD8_C10_MKI67 and Treg_C3_MKI67 (Supplementary Fig. 2c and Supplementary Data 3). The co-expression of

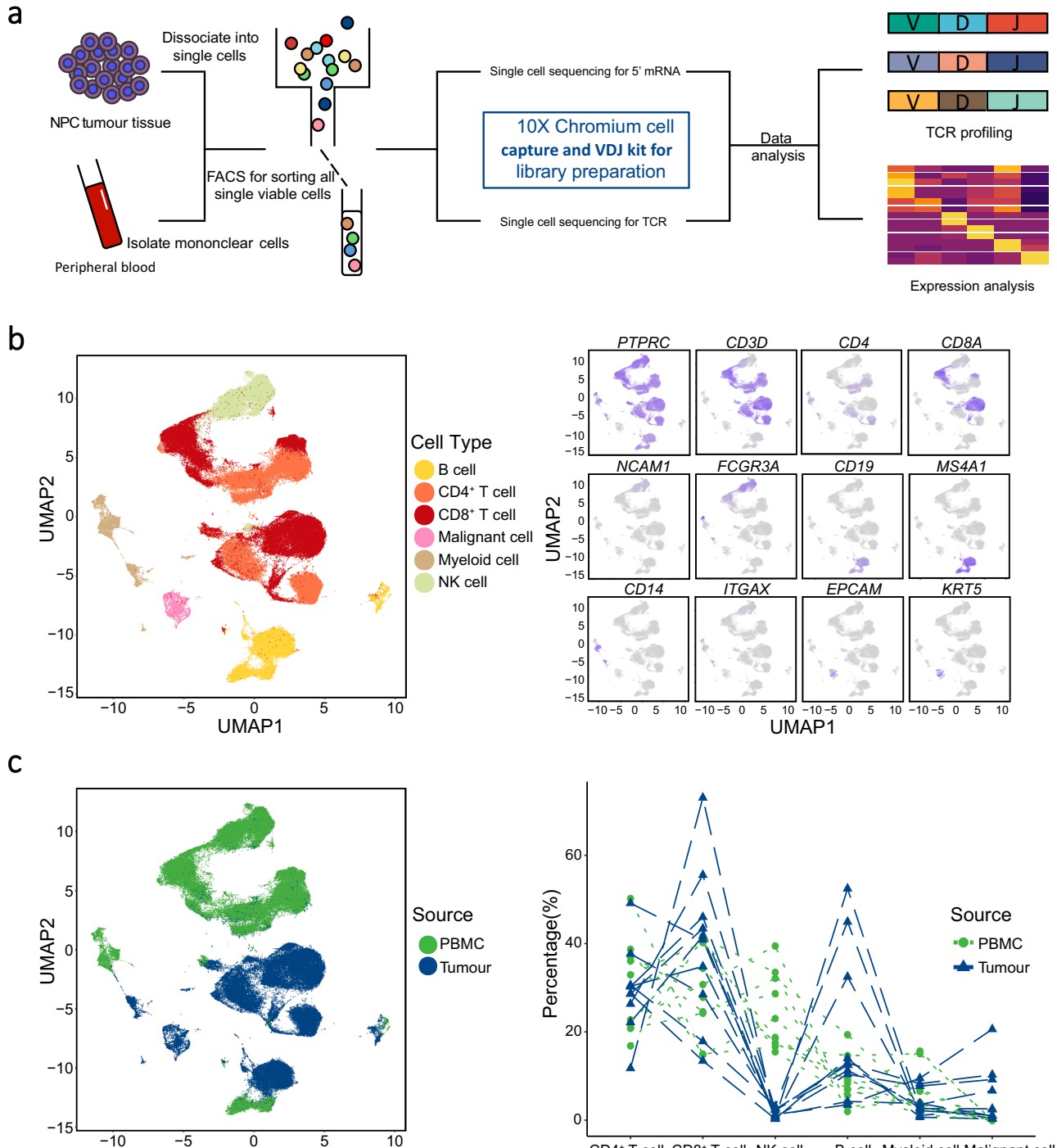

**Fig. 1 The landscape profiling of single cells in NPC tumours and matching PBMC. a** An experimental scheme diagram highlights the overall study design. Single viable cells were collected using flow cytometry sorting (FACS) and subjected for cell barcoding. The cDNA libraries of 5'-mRNA expression and TCR were constructed independently, followed by high throughput sequencing and downstream analyses. **b** UMAP plot of 176,447 single cells grouped into six major cell types (left panel) and the normalised expression of marker genes for each cell type (right panel). Each dot represents one single cell, coloured according to cell type (left panel), and the depth of colour from grey to blue represents low to high expression (right panel). **c** UMAP plot of the above single cells coloured according to their origins from peripheral blood or tumour (left panel), and the fraction of cell types originating from each patient (right panel). Each dot represents one single cell, coloured according to sample origin.

exhaustion and effector genes in tumour infiltrating T cells has been also demonstrated recently in NPC[17]. Together, these observations suggest that T cells exhibited anti-tumour effects against antigens, but their effector functions were somehow suppressed in the TME of NPC. By contrast, we observed naïve

gene signatures (high expression of *TCF7*, *SELL*, *CCR7* and *LEF1*) in the resting T cells in PBMC, including CD4_C1_LEF1, CD8_C1_LEF1, CD8_C2_TCF7, and DN_LEF1 (CD4⁻CD8⁻) cell clusters, and partially in Treg_C1_SELL and DP_TCF7 (CD4⁺CD8⁺; Supplementary Fig. 2b). Besides, we identified two

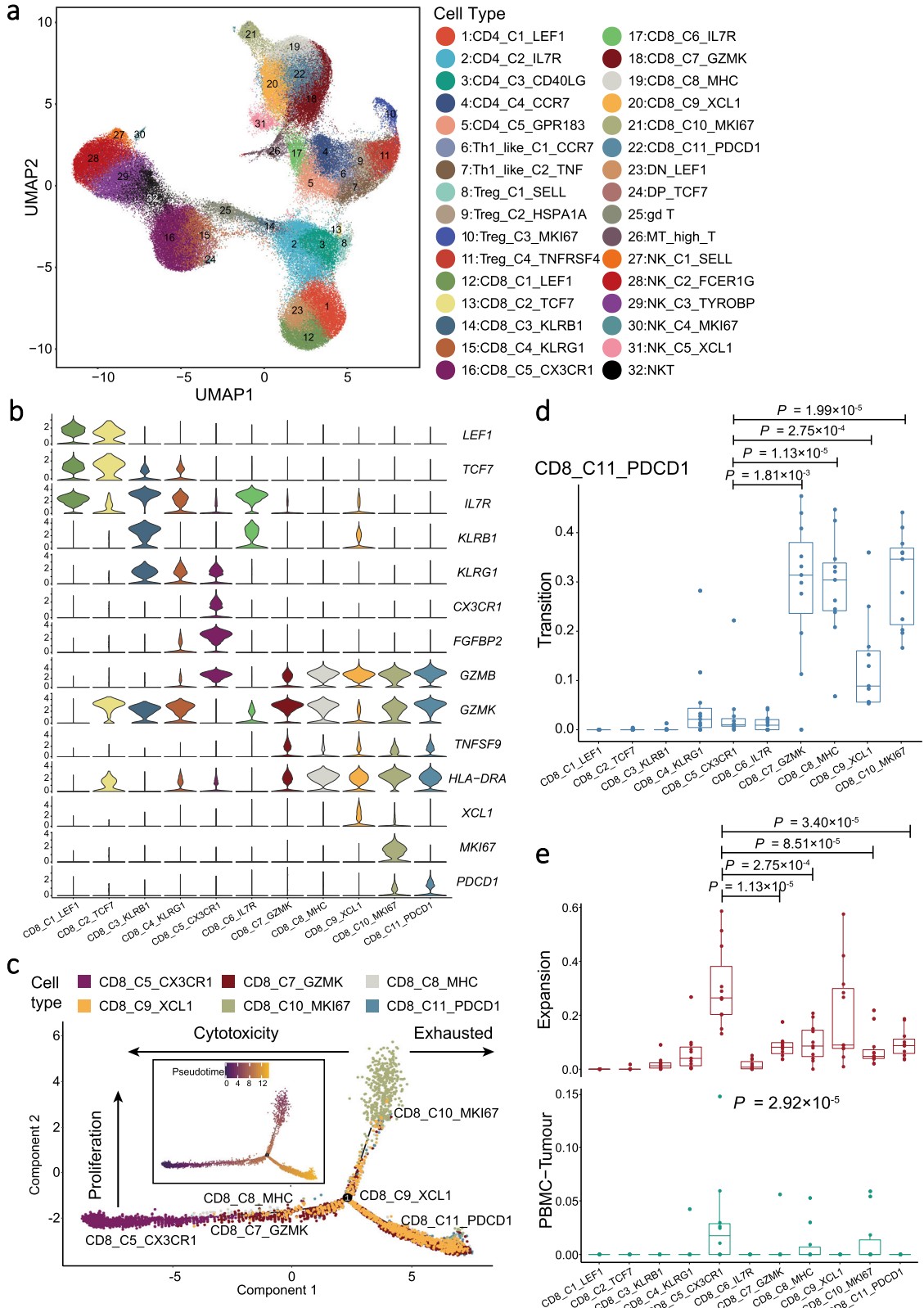

clusters of CD4+ Th1-like cells in tumours, including Th1_like_C1_CCR7 and Th1_like_C2_TNF, with specific expression of naïve T cell markers and pro-inflammatory cytokines, respectively, as well as a common expression of Th1-like cell markers[21] (*CXCL13*, *BHLHE40*, and *CXCR3*; Supplementary Fig. 2d).

Next, we performed T cell receptor (TCR) repertoire analysis based on the sequences of α and β chains of TCR, which revealed

38,720 (32.97%; out of 117,447) T cells with detectable TCR α-β pairs or clonotypes after strict quality control (Supplementary Fig. 2e and Supplementary Data 4). We observed no sharing of any identical TCR clonotype among different patients with NPC, although they had certain preferences of V and J fragments as well as V–J pairs (Supplementary Fig. 3a, b). Interestingly, we observed the sharing of the most variable CDR3 sequences across

**Fig. 2 Expression profile and development of CD8$^+$ T cells. a** UMAP plot of 141,875 T and NK cells grouped into 32 cell types. Each dot represents a cell, coloured according to the cell types indicated at the right legend. **b** Violin plots showed the normalised expression of CD8$^+$ T cell markers (rows) in each CD8$^+$ T cell cluster (columns). Cell clusters and the expression level of a gene are indicated at the x- and y-axis, respectively. **c** Pseudotime trajectory analysis of selected CD8$^+$ T cells (CD8_C5, CD8_C7, CD8_C8, CD8_C9, CD8_C10, and CD8_C11; $n = 10,000$) with high variable genes. Each dot represents one single cell, coloured according to its cluster label. The inlet plot showed each cell with a pseudotime score from dark blue to yellow, indicating early and terminal states, respectively. For CD8$^+$ T cell clusters, 10,000 cells were randomly selected for the analysis. **d** Box plots showed the transition-index scores of exhausted CD8$^+$ T cells (CD8_C11_PDCD1) and other CD8$^+$ T cells ($n = 10$). Comparison was made using a two-sided Wilcoxon test. Cell clusters and transition-index scores are indicated at the x- and y-axis, respectively. Endpoints depict minimum and maximum values; centre lines denote median values; whiskers denote 1.5× the interquartile range; coloured dots denote each patient. **e** Box plots showed the expansion- (top panel) and PBMC-Tumour migration-index (bottom panel) scores of each CD8$^+$ T cell cluster ($n = 10$). Each comparison was made using either a two-sided Wilcoxon test (top panel) or Kruskal–Wallis test (bottom panel). Cell clusters are indicated at the x-axis, and the y-axis shows the expansion- and PBMC-Tumour migration-index scores at the top and bottom panel, respectively. Endpoints depict minimum and maximum values; centre lines denote median values; whiskers denote 1.5× the interquartile range; coloured dots denote each patient.

the patient samples (Supplementary Fig. 3b), among which CAVRGTGTASKLTF and CASSFSGANVLTF have been associated with the recognition of MLANA and EBV antigens in the VDJ database[22], respectively (Supplementary Table 4). Moreover, we observed that both CD4$^+$ and CD8$^+$ T cells have more clonal T cells, which are derived from identical TCR clonotypes and consistent with a previous study[17], in the tumours compared to the PBMC, suggesting the clonal expansion of certain dominant clones of tumour infiltrating T cells upon continuous stimulations by tumour antigens (Supplementary Fig. 3c).

**Diversity of CD8$^+$ T cells and the development of exhausted intratumoral CD8$^+$ T cells.** We identified a total of 62,244 CD8$^+$ T cells in all NPC samples, which were grouped into 11 clusters based on their expression of canonical markers, including two naïve (CD8_C1_LEF1 and CD8_C2_TCF7), blood central memory (CD8_C3_KLRB1), blood effector memory (CD8_C4_KLRG1), high migration (CD8_C5_CX3CR1), tumour central memory (CD8_C6_IL7R), tumour effector memory (CD8_C7_GZMK and CD8_C8_MHC), tissue resident memory (CD8_C9_XCL1), high proliferating (CD8_C10_MKI67), and exhausted (CD8_C11_PDCD1) T cells (Fig. 2a, b, Supplementary Data 1 and 2). The majority (>97.68%) of CD8_C6_IL7R, CD8_C7_GZMK, CD8_C8_MHC, CD8_C9_XCL, CD8_C10_MKI67, and CD8_C11_PDCD1 were found in NPC tumours, whereas the majority (>94.85%) of CD8_C1_LEF1, CD8_C2_TCF7, CD8_C3_KLRB1, CD8_C4_KLRG1, and CD8_C5_CX3CR1 were in the peripheral blood (Supplementary Data 1).

To evaluate the functional status of CD8$^+$ T cells, we calculated cytotoxicity, proliferation, and exhaustion scores for all CD8$^+$ T cell clusters. We observed the highest cytotoxicity score for CD8_C5_CX3CR1, the highest proliferation score for CD8_C10_MKI67, and the highest exhaustion score for CD8_C11_PDCD1 (Supplementary Fig. 2c and 4a), suggesting their potential cytotoxic, proliferation, and exhausted functions, respectively. Next, the DEG analysis revealed high expression of chemokine receptors (CX3CR1, CXCR1, and CXCR2), S1P receptors (S1PR1, S1PR4, and S1PR5), and integrins (ITGB2, ITGA4, ITGAL, and ITGB7) in CD8_C5_CX3CR1, which were responsible for the regulation of CD8$^+$ T cell migration[23] (Supplementary Fig. 4b). Moreover, signalling pathway enrichment analyses of the genes with differential expression revealed that tumour cytotoxic CD8$^+$ T cell clusters (CD8_C7_GZMK, CD8_C8_MHC, and CD8_C9_XCL1) were enriched with the pathways related to cytokine production and lymphocyte activation; and CD8_C5_CX3CR1 was enriched with the pathways related to leukocyte trans-endothelial migration and leukocyte migration (Supplementary Fig. 4c), which are

consistent with their capability in peripheral circulation and infiltrating to tumour[24].

To further explore the development of CD8$^+$ T cells in NPC, we first performed pseudotime trajectory analysis using Monocle2 to order each CD8$^+$ T cell along trajectories according to their expression and transition profiles. We observed the developmental trajectories from CD8_C5_CX3CR1 cells at the initial state or CD8_C10_MKI67 cells at the intermediate state to CD8_C11_PDCD1 cells at the terminal state (Fig. 2c). Supportively, compared to the exhausted CD8_C11_PDCD1, CD8_C10_MKI67 had an intermediate exhaustion score (Supplementary Fig. 4a) and lower expression of known exhaustion markers including PDCD1, LAG3, and HAVCR2 (Supplementary Fig. 2b). TCR repertoire sequencing revealed 21,099 CD8$^+$ T cells (out of 62,244) with TCR clonotypes (Supplementary Fig. 4d and Supplementary Data 4). We observed that CD8_C11_PDCD1 shared considerable proportions of identical TCRs with other CD8$^+$ T cell clusters, ranging from 17.68% to 41.67% for infiltrating T cell clusters and 5.31% for peripheral CD8_C5_CX3CR1 (Supplementary Fig. 4e). To track the dynamic relationships among T cell clusters from NPC tumour and peripheral blood, we quantitated the expansion (exp, clonal expansion), migration (migr) and transition (tran, developmental transition or differentiation) of T cells using gene expression and TCR information with STARTRAC method[24]. Consistently, we observed the highest transition mobility of CD8_C11_PDCD1 with CD8_C10_MKI67, followed by CD8_C7_GZMK, CD8_C8_MHC, and CD8_C9_XCL1 (Fig. 2d). These observations strongly suggest that CD8_C11_PDCD1 cells were mainly expanded by proliferating pre-exhausted intratumoral CD8$^+$ T cells. Moreover, we observed that CD8_C5_CX3CR1 had the largest number of clonal T cells (Supplementary Fig 4d) and the highest expansion mobility in CD8$^+$ T cell clusters (Fig. 2e). Furthermore, CD8_C5_CX3CR1 had the highest proportion of shared TCR between peripheral blood and tumour (Fig. 2e). The proportions of TCRs shared with peripheral CD8_C5_CX3CR1 ranged from 4.76% to 12.77% for infiltrating CD8$^+$ T cell clusters (Supplementary Fig. 4e). These data suggest a common origin of the intratumoral CD8$^+$ T cells in NPC tumour from CD8$^+$ T cells in peripheral blood including CD8_C5_CX3CR1 cells.

**The diversity and trajectory of Treg cells in NPC.** Treg cells are potent suppressors of immune cells and are essential to maintaining immunological tolerance and homoeostasis. We identified a total of 11,631 Treg cells based on their transcription of canonical markers (CD4, IL2RA, and FOXP3), which were grouped into four cell clusters, including Treg_C1_SELL, Treg_C2_HSPA1A, Treg_C3_MKI67, and Treg_C4_TNFRSF4 (Fig. 2a, Supplementary Fig. 5a, Supplementary Data 1 and 2). We observed that the proportion of Treg cells among CD4$^+$

T cells in the tumours was much higher than that in the PBMC (Supplementary Fig. 5b). All Treg_C4_TNFRSF4 cells and the majority of Treg_C2_HSPA1A (99.5%; 4,762 out of 4,786) and Treg_C3_MKI67 (90.0%; 1,187 out of 1,319) cells were found in the tumours, while Treg_C1_SELL cells were in the PBMC (Supplementary Data 1). To explore the immune-regulatory functions of Treg cells, we first calculated the IL2R scores for each cell based on their expression level of *CD25* (IL2RA), *CD122* (IL2RB), and *CD132* (IL2RG) using the AddModuleScore function implemented in Seurat software. The three genes encode transmembrane proteins that form a receptor complex competitively binding IL2 (the T cell growth factor) with high affinity so as to inhibit effector T cells[25]. We observed the highest IL2R score for Treg_C4_TNFRSF4 among all Treg clusters (Fig. 3a), suggesting the strongest IL-2 binding potential of Treg_C4_TNFRSF4 cells. Similarly, we also observed the highest inhibitory and co-stimulatory scores for Treg_C4_TNFRSF4 cells based on their expression levels of genes with immune-inhibitory functions and co-stimulatory receptors, respectively (Fig. 3a and Supplementary Fig. 5a), suggesting that Treg_C4_TNFRSF4 cells had a stronger suppression potential on immune response and were much activated than the other Treg cells. Consistently, such a subset of Treg cells were also identified in CRC, NSCLC, and hepatocellular carcinoma (HCC), with high activation and immune-suppressive potential as indicated by the high IL2R, inhibitory, and co-stimulatory scores (Supplementary Fig. 5c). Besides, we observed elevated expression levels of chemokine receptors in Treg_C4_TNFRSF4 cells, including *CXCR3*, *CXCR6*, and *CCR8* that have been implicated in several cancers[26] (Supplementary Fig. 5a).

To characterise the potential functions of Treg cells, we first performed signalling pathway enrichment analyses for each Treg cluster based on the expression levels of genes implicated in each pathway. We observed a distinct pattern of pathway enrichment for each Treg cluster, suggesting their various functions. Particularly, the 'cytokine-cytokine receptor interaction' was highly enriched in Treg_C4_TNFRSF4 (Fig. 3b), consistent with their chemotactic potentials as mentioned earlier. Moreover, the 'interleukin-10 production', 'TNF signalling pathway' and 'NF-κB signalling pathway' were enriched in both Treg_C4_TNFRSF4 and Treg_C2_HSPA1A (Fig. 3b). As such, we further compared the major pathways between Treg_C4_TNFRSF4 and Treg_C2_HSPA1A using Gene Set Enrichment Analysis (GSEA), which revealed a higher enrichment of pathways related to cell cycle, chemokine, TGF-β, and negative regulation of T cell proliferation, as well as transcription factors activating NF-κB and STAT pathways in Treg_C4_TNFRSF4 (Supplementary Fig. 5d). Since Treg_C4_TNFRSF4 expressed *CCR8* specifically (Supplementary Fig. 5a), we used the normalised mRNA ratio of *CCR8/FOXP3* to estimate the fraction of Treg_C4_TNFRSF4 cells in Treg cells (*FOXP3*+). Survival analysis showed that the higher ratio of *CCR8/FOXP3* was associated with the decreased progression-free survival (PFS; Supplementary Fig. 5e), suggesting that a higher fraction of Treg_C4_TNFRSF4 cells with activated potential in Treg cells had a strong immune-suppressive function in NPC.

To trace the origin of intratumoral Treg cells, we first performed pseudotime trajectory analysis using Monocle2, which revealed the most terminal status with the highest pseudotime scores for Treg_C4_TNFRSF4 cells and two developmental trajectories of Treg_C4_TNFRSF4 cells from Treg_C1_SELL cells in PBMC and Treg_C3_MKI67 cells in tumours (Fig. 3c). We next examined the expression of tissue resident markers (*CD69*, *ITGAE*, and *BHLHE40*; Supplementary Table 5) in Treg cells and observed much higher expression of *ITGAE* and *BHLHE40* in Treg_C4_TNFRSF4 and Treg_C3_MKI67 cells than

Treg_C2_HSPA1A and Treg_C1_SELL cells (Supplementary Fig. 6a). Given that the majority of Treg_C2_HSPA1A cells was in tumours and originated from Treg_C1_SELL cells according to the pseudotime trajectory analysis (Fig. 3c), the scarce expression of the resident markers might suggest the most recent recruitment of Treg_C2_HSPA1A cells from peripheral blood.

TCR repertoire analysis revealed 17,621 (out of 47,384) CD4+ T cells assigned with clonotypes, among which Treg_C2_HSPA1A and Treg_C4_TNFRSF4 had intermediate numbers of clonotypes (Supplementary Fig. 6b and Supplementary Data 4). Noteworthily, Treg_C4_TNFRSF4 had the largest proportion of clonal cells, meaning the highest clonality, among all CD4+ T cells (Supplementary Fig. 6b). Consistently, we observed that Treg_C4_TNFRSF4 had the highest expansion score, meaning the highest degree of clonal expansion, among the Treg cell clusters (Fig. 3d). We also observed the highest migration score, meaning the highest mobility, for Treg_C1_SELL derived from the PBMC (Fig. 3d). DEG analysis revealed that Treg_C1_SELL had high expression of chemokine receptors *CCR4*, which are chemotactic counterparts for *CCL5*, *CCL17*, and *CCL22* produced by intratumoral CD8+ T, NK, and myeloid cells in NPC (Supplementary Figs. 5a and 6c). These observations suggest that the migration capability and chemotactic interaction potentials with intratumoral cells make the movement of peripheral Treg_C1_SELL cells to tumour site possible. Indeed, we observed a small number of shared TCRs between Treg cells from tumour and peripheral blood (Supplementary Fig. 6d), which is consistent with a previous finding that intratumoral Treg cells were partially recruited from peripheral blood[27]. We further examined the transition mobility of Treg_C2_HSPA1A and Treg_C4_TNFRSF4 with other Treg cells. We observed that Treg_C4_TNFRSF4 cells had the highest transition mobility with Treg_C3_MKI67 cells, followed by Treg_C2_HSPA1A and Treg_C1_SELL cells; and Treg_C2_HSPA1A cells had high transition mobility with Treg_C4_TNFRSF4 and Treg_C3_MKI67 cells (Fig. 3e). These observations again supported the developmental trajectory of intratumoral Treg_C4_TNFRSF4 cells from naïve Treg_C1_SELL cells through intermediate Treg_C2_HSPA1A or Treg_C3_MKI67 cells (Fig. 3c).

**Diversity of B cells in NPC.** We identified a total of 22,892 B cells, which were grouped into nine clusters (Supplementary Fig. 7a, Supplementary Data 1 and 2). Among them, B_C1_TCL1A, B_C2_FCRL3, and Plamsa_C1_IgA clusters were derived from PBMC and the other six clusters were from tumour samples (Supplementary Fig. 7a). DEG analysis revealed unique gene signatures for B cell clusters in tumour samples, including B_C5_ISG15 with interferon induced genes, B_C6_HSPA1A with stressful gene expression, and Plasma_C2_IgG with elevated expression level of IgH genes (Supplementary Fig. 7b). We further identified two B cell clusters (B_C1_TCL1A and B_C4_IFITM3) before the class switch recombination based on the expression of *IGHM* and *IGHD* (Supplementary Fig. 7b and Supplementary Table 5). Moreover, correlation analysis revealed that the expression of *TCL1A* was highly correlated with that of *IGHM* and *IGHD* in the two clusters, which could be a sufficient marker to classify B cells before the class switch recombination (Supplementary Fig. 7c). Signalling pathway enrichment analyses of the genes with differential expression revealed that B cell clusters were enriched with various pathways related to immune regulation (Supplementary Fig. 7d). Particularly, B_C4_IFITM3, B_C5_ISG15, and B_C6_HSPA1A had the enrichment of 'EBV infection', 'defence response to virus', 'viral carcinogenesis', and 'response to interferon-gamma' pathways, suggesting that the

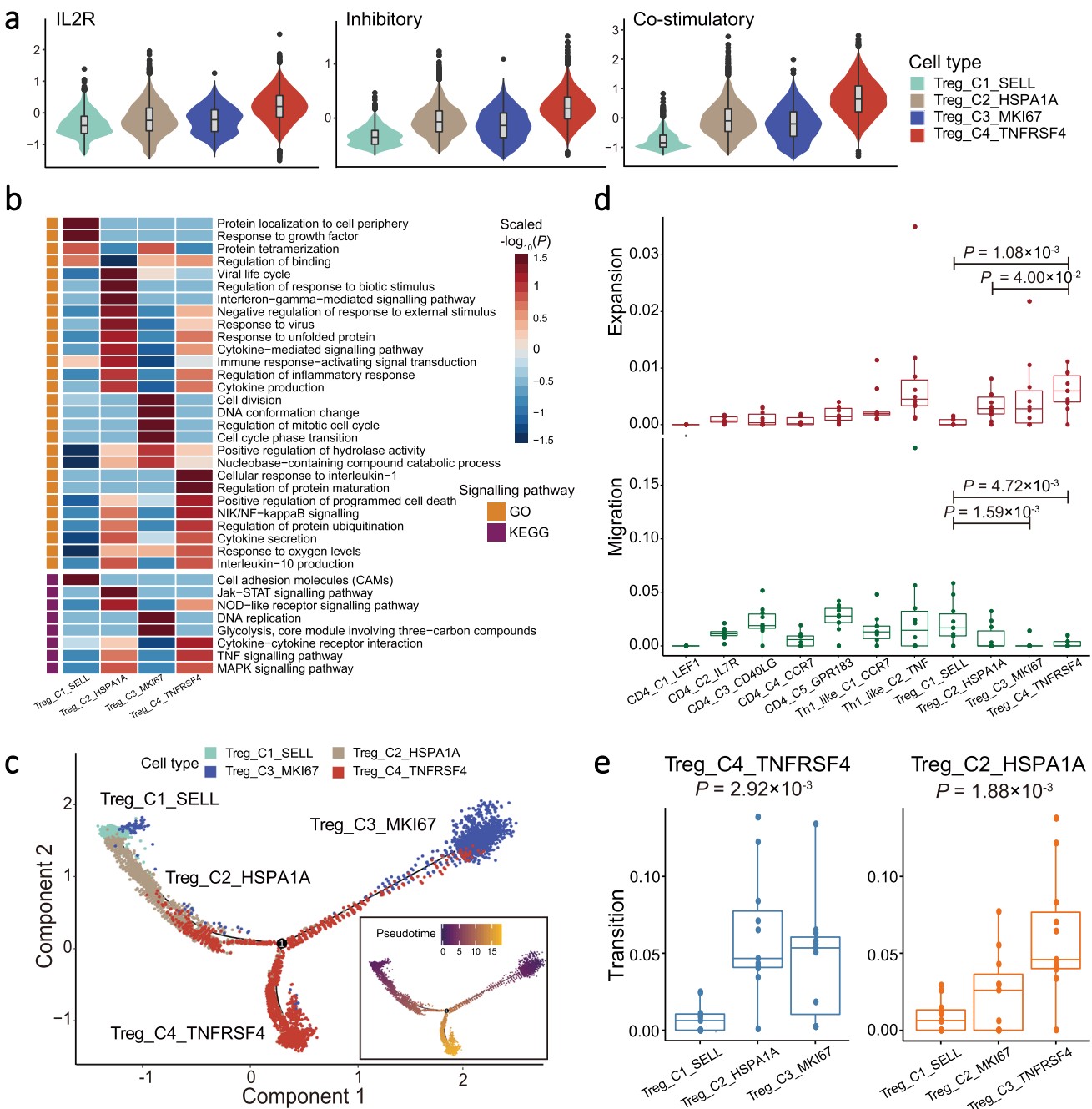

**Fig. 3 Expression profile and development of Treg cells. a** Violin plots showed the IL2R (left panel), inhibitory (middle panel), and co-stimulatory (right panel) scores for each Treg cell cluster ($n = 11,631$). Box plots inside the violins indicated the quartiles of corresponding score levels. Endpoints depict minimum and maximum values; centre lines denote median values; whiskers denote 1.5× the interquartile range; black dots denote each cell. Violin plots are coloured according to cell types, and signature scores are indicated at the y-axis. **b** Heatmap showed the selected signalling pathways (rows) that were significantly enriched in GO and KEGG analyses for each Treg cell cluster (columns). Filled colours from blue to red represent scaled expression levels (normalised $-\log_{10}P$ values) from low to high. $P$ values were calculated by one-sided hypergeometric test and adjusted for multiple comparisons. Orange and purple squares on the left column represent the results derived from GO and KEGG signalling pathways analysis, respectively. **c** Pseudotime trajectory analysis of Treg cells (Treg_C1, Treg_C2, Treg_C3, and Treg_C4; $n = 11,631$) with high variable genes. Each dot represents one single cell, coloured according to its cluster label. The inlet plot showed each cell with a pseudotime score from dark blue to yellow, indicating early and terminal states, respectively. **d** Box plots showed the expansion- (top panel) and migration-index (bottom panel) scores of each CD4$^+$ T cell cluster ($n = 10$). Comparison was made using two-sided Wilcoxon test. Cell clusters are indicated at the x-axis, and the y-axis shows the expansion- or migration-index at the top or bottom panel, respectively. Endpoints depict minimum and maximum values; centre lines denote median values; whiskers denote 1.5× the interquartile range; coloured dots denote each patient. **e** Box plots showed the transition-index scores of Treg_C4_TNFRSF4 (left panel) and Treg_C2_HSPA1A (right panel) with other Treg cells ($n = 10$). Comparison was made using two-sided Kruskal-Wallis test. Cell clusters and the transition-index scores are indicated at the x- and y-axis, respectively. Endpoints depict minimum and maximum values; centre lines denote median values; whiskers denote 1.5× the interquartile range; coloured dots denote each patient.

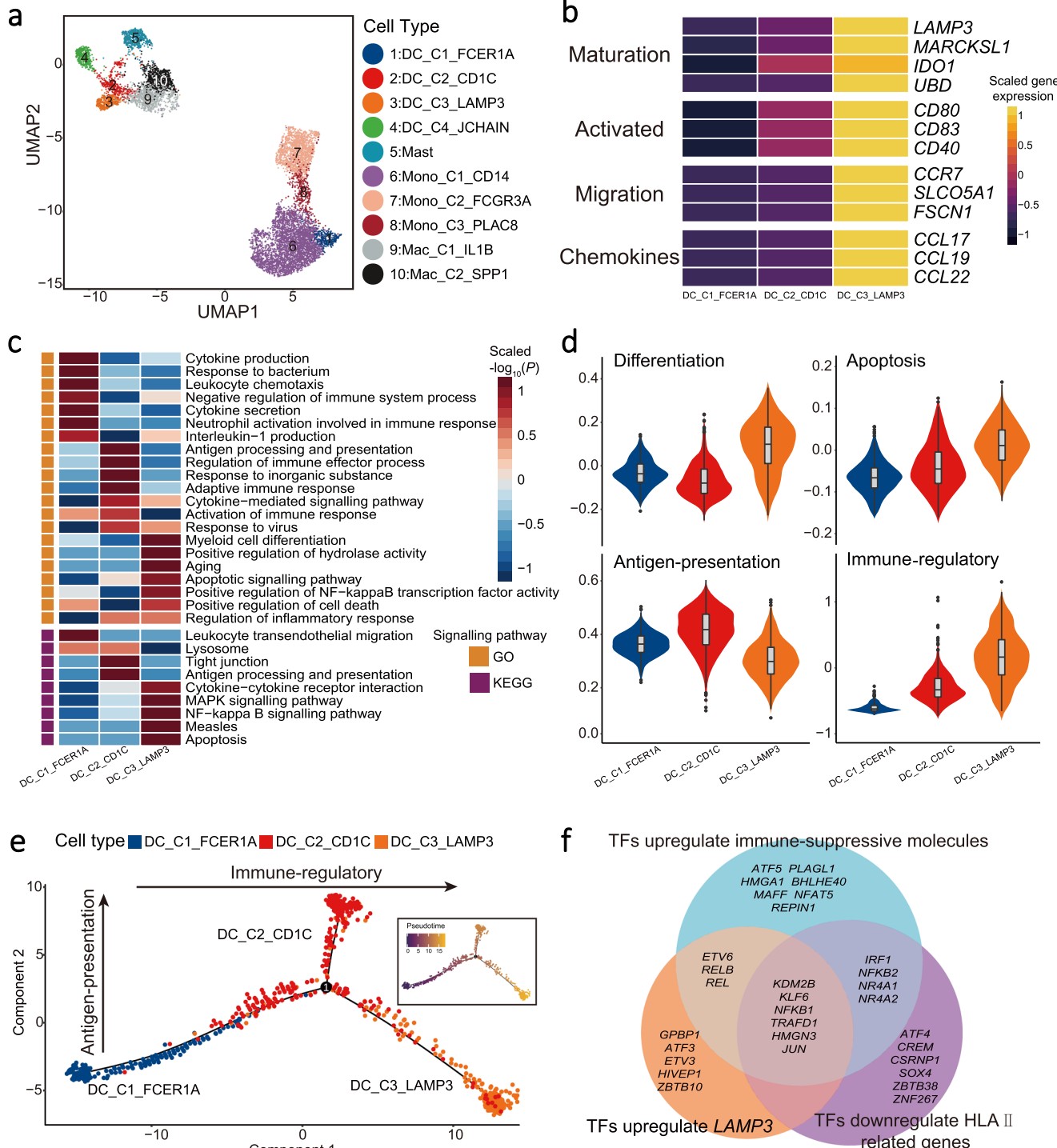

three groups of cells might be responsible for the immune response against EBV infection (Supplementary Fig. 7d).

**Tumour-associated LAMP3+ DCs display a tolerogenic phenotype in NPC.** A total of 8,893 myeloid cells were identified and clustered into 10 subsets including one for mast cells, five for monocyte or macrophage cells, three for conventional dendritic cells, and one for plasmacytoid dendritic cells (Fig. 4a, Supplementary Fig. 8a and 8b, Supplementary Data 1 and 2). Among the four clusters of dendritic cells, DC_C2_CD1C, DC_C3_LAMP3, and DC_C4_JCHAIN were derived from tumours, and DC_C1_FCER1A was derived from peripheral blood, which was

assigned as monocyte-like DC because of the expression of monocyte marker *S100A8* (Supplementary Fig. 8a, b). Noteworthily, we identified DC_C3_LAMP3 cells as a group of DCs with high maturation, activation, and migration potentials in NPC, based on the expression levels of the signature genes related to maturation (*LAMP3, MARCKSL1, IDO1,* and *UBD*), activation (*CD80, CD83,* and *CD40*), and migration (*CCR7, FSCN1,* and *SLCO5A1*; Fig. 4b, Supplementary Fig. 8c and Supplementary Table 5), respectively. Moreover, DC_C3_LAMP3 cells had high expression of special chemokine ligands (*CCL17, CCL19,* and *CCL22*), which are known to recruit immune cells expressing chemokine receptors *CCR4, CCR7,* and *CXCR3* (Fig. 4b). We also

**Fig. 4 Expression and development of dendritic cells. a** UMAP plot of 8,893 myeloid cells grouped into 10 cell types. Each dot represents a cell, coloured according to cell types. **b** Heatmap showed the normalised mean expression of genes associated with maturation, activation, migration, and chemokine ligand (rows) in three dendritic cell clusters (DC_C1, DC_C2, and DC_C3; columns). Filled colours from black to yellow represent scaled gene expression levels from low to high. **c** Heatmap showed the selected signalling pathways (rows) with significant enrichment of GO and KEGG terms for three dendritic cell clusters (DC_C1, DC_C2, and DC_C3; columns). Filled colours from blue to red represent scaled expression levels (normalised $-\log_{10}P$ values) from low to high. *P*-values were calculated by one-sided hypergeometric test and adjusted for multiple comparisons. Orange and purple squares on the left column represent the results derived from GO and KEGG signalling pathways analysis, respectively. **d** Violin plots showed the differentiation, apoptosis, antigen presentation, and dysfunction scores of three dendritic cell cluster (DC_C1, DC_C2, and DC_C3; *n* = 1134). Box plots inside the violins indicated the quartiles of corresponding score levels. Endpoints depict minimum and maximum values; centre lines denote median values; whiskers denote 1.5 × the interquartile range; black dots denote each cell. Cell clusters and the signature scores are indicated at the x- and y-axis, respectively. **e** Pseudotime trajectory analysis of three dendritic cell clusters (DC_C1, DC_C2, and DC_C3; *n* = 1134) with high variable genes. Each dot represents one single cell, coloured according to its cluster label. The inlet plot showed each cell with a pseudotime score from dark blue to yellow, indicating early and terminal states, respectively. **f** Venn diagram showed overlapped transcription factors regulating *LAMP3* gene, immune-suppressive molecules, and HLA-II in DC_C3_LAMP3 cells.

---

observed significant correlations of expression between the marker gene *LAMP3* and other functional genes related to maturation, migration, activation, and chemokine ligands in DC_C3_LAMP3 (Supplementary Fig. 8d). These observations suggest that DC_C3_LAMP3 cells might be LAMP3⁺ DCs, which are featured with high migration, activation, and maturation in several cancers as reported previously[28,29].

Signalling pathway enrichment analyses using GO and KEGG revealed a specific pattern of enriched pathways among the three conventional DC cell clusters, where the 'antigen processing and presentation' was significantly upregulated in DC_C2_CD1C but downregulated in DC_C3_LAMP3 (Fig. 4c). Moreover, apoptosis, NF-κB, and MAPK signalling pathways as well as myeloid cell differentiation were also upregulated in DC_C3_LAMP3 compared to other two clusters (Fig. 4c). These observations are consistent with the GSEA analyses (Supplementary Fig. 8e). As such, we scored the expression levels of genes related to these pathways in each cluster (Supplementary Data 3), which revealed the highest levels of differentiation and apoptosis but the lowest antigen presentation for DC_C3_LAMP3 (Fig. 4d). Furthermore, we observed that the gene signatures corresponding to the activation of immune response was reduced in DC_C3_LAMP3 (Fig. 4c), which was consistent with the highest immune-regulatory score and the increased expression of a subset of immune-suppressive genes, including *CD274* (PD-L1), *PDCD1LG2* (PD-L2), *CD200*, *EBI3*, *IDO1*, *IL4I1*, *SOCS1*, *SOCS2*, and *SOCS3* (Supplementary Fig. 9a and Supplementary Data 3). Besides, we observed a similar expression profile of LAMP3⁺ DC among NPC, HCC, and NSCLC (Supplementary Fig. 9a). These observations suggest that DC_C3_LAMP3 cells could be considered as a group of regulatory and tolerogenic DCs, which restrain the activation of T cells[30].

Next, we performed pseudotime trajectory analysis and observed that DC_C1_FCER1A cells developed into two branches including DC_C2_CD1C and DC_C3_LAMP3 cells, and DC_C3_LAMP3 cells had the highest pseudotime score meaning the most differentiated and matured DC (Fig. 4e). Combined with their immune-regulatory and antigen-presenting scores (Fig. 4d), these data suggest that DC_C1_FCER1A cells in peripheral blood might infiltrate to tumour, convert to DC_C2_CD1C cells with increased antigen-presenting capacity and to immune-suppressive DC_C3_LAMP3 cells (Fig. 4e). Consistently, we observed the similar pattern of changes in the expression of transcription factors (TFs) specific for the genes with differential expression from DC_C1_FCER1A to DC_C2_CD1C and then DC_C3_LAMP3 cells (Supplementary Fig. 9b). We further constructed an accurate cellular network to infer the regulons associated with transcription factors and signalling molecules in DC_C3_LAMP3 using ARACNe (Supplementary Data 5). We

observed that the upregulation of *LAMP3* was linked with multiple TFs, including *ETV3*, *ETV6*, *HMGN3*, *GPBP1*, *TRAFD1*, *ATF3*, *KDM2B*, *JUN*, *HIVEP1*, *KLF6*, *ZBTB10*, and *NFKB1* that have been related to the maturation of DC in mouse[31], while the downregulation of *LAMP3* was linked with *CREM* (Supplementary Fig. 9c). We also observed that among these TFs *KDM2B*, *KLF6*, *ETV6*, *JUN*, *HMGN3*, and *TRAFD1*, as well as *NFKB1*, *REL*, and *RELB* in the NF-κB pathway were linked with the upregulated expression of immune-suppressive molecules like *CD274*, *PDCD1LG2*, *CD200*, and *IDO1*, but the downregulated expression of HLA-class II genes (Fig. 4f and Supplementary Fig. 9c). By contrast, *SOX4* and *CREM* were associated with the downregulation of *CD274*, *CD200*, and *IDO1* (Supplementary Fig. 9c). These observations suggest that multiple TFs regulate the immune-suppressive function, antigen-presenting capacity, and maturation of DC_C3_LAMP3 in NPC.

**Heterogeneity of malignant cells with different EBV infection status.** We identified a total of 2,787 malignant epithelial cells in NPC tumours based on their presence of large-scale chromosomal copy number variation (CNV) compared to a reference data of normal epithelial cells[32] (see Methods; Fig. 5a). Given that EBV is a known factor responsible for the malignant transformation and tumorigenesis of NPC[33], we examined the expression of EBV molecules in the malignant cells and divided them into EBV⁺ (EP_C1_LMP1) and EBV⁻ (EP_C2_EPCAM) malignant cells according to their detectable or not EBV transcripts (*LMP-1/ BNLF2a/b*, *RPMS1/A73*, *LMP-2A/B*, and *BNRF1*; Fig. 5b, c, Supplementary Fig. 10a, Supplementary Data 1 and 2). We observed higher expression of *EPHA2* and *EGFR* in EP_C1_LMP1 cells (Fig. 5d), which have been related to the susceptibility of EBV infection[34]. Moreover, immunofluorescence staining of an EBV-encoded protein (LMP1) confirmed the presence of EBV⁺ malignant cells (LMP1⁺EPCAM⁺) and EBV⁻ malignant cells (LMP1⁻EPCAM⁺) in NPC (Fig. 5e). We also observed specifically high activations of the major genes involved in NF-κB and Notch pathways as well as chemokines including *CX3CL1* in EP_C1_LMP1 compared to EP_C2_EPCAM cells (Fig. 5d). Consistently, we observed high expression of *CX3CL1* in an independent collection of NPC tumours (*n* = 113) compared to non-cancerous samples (rhinitis, *n* = 10; Supplementary Fig. 10b). Interestingly, we noted that the overexpression of *CX3CR1*, the receptor of *CX3CL1*, in multiple types of immune cells in peripheral blood (Supplementary Fig. 10c). Signalling pathway enrichment analyses revealed that EP_C1_LMP1 were enriched with cytokine-mediated, regulation of cell death, apoptosis, and cancer-related pathways (Fig. 5f). Taken together, these observations suggest that malignant NPC cells exhibit different

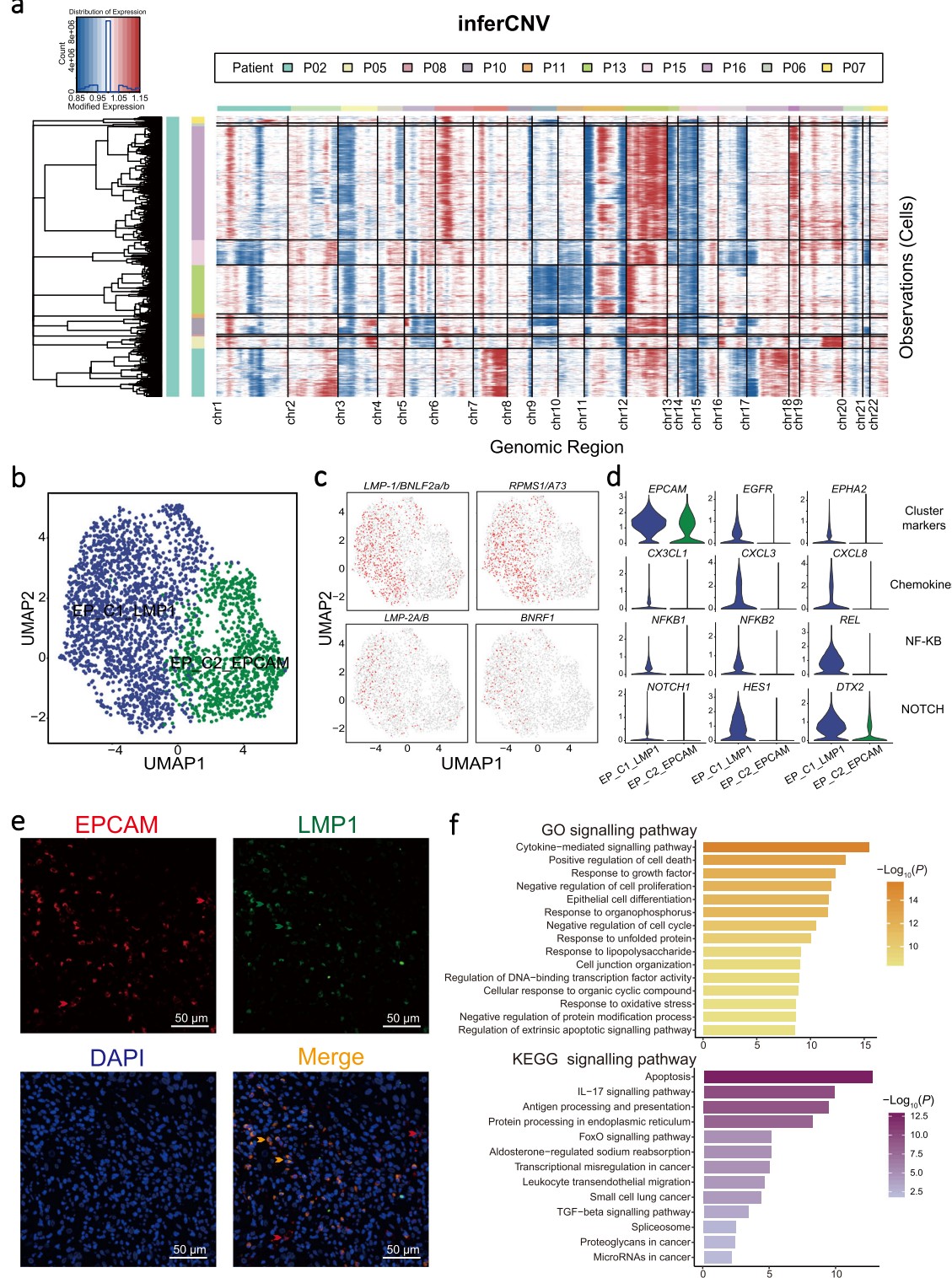

susceptibility to EBV infection, leading to distinct expression profiles.

To further explore the intra-tumour and inter-tumour heterogeneity of malignant NPC cells, we first divided them into five prominent cell subgroups (C1–C5; Supplementary Fig. 10d, e), using clustering analysis without EBV information. Next, we deciphered the variations of gene expression in malignant cells for different clusters using gene set variation analysis (GSVA), which revealed a distinct enrichment of signalling pathway for each cluster (Supplementary Fig. 10f). We observed variable proportions of cell subtypes among different tumour samples (Supplementary Fig. 10e), which might contribute to the inter-tumour heterogeneous expression profiles. Indeed, the GSVA data showed different enrichments of signalling pathways among

**Fig. 5 Heterogeneity of malignant cells with distinct EBV infection in tumour tissues. a** Heatmap showed the large-scale CNVs for epithelial cells (rows along y-axis) from 10 NPC tumours. CNVs were inferred according to the average expression of 100 genes spanning each chromosomal position (x-axis). Red: gains; blue: losses. Malignant NPC cells from different patients and the range of different chromosomes are indicated as different colour bars on the left and top to the heatmap, respectively. **b** UMAP plot of 2,787 malignant cells grouped into two cell clusters (EP_C1_LAMP1 and EP_C2_EPCAM). Each dot represents a cell, coloured according to a cell cluster. **c** UMAP plots showed the expression of EBV-encoded genes (LMP-1/BNLF2a/b, RPMS1/A73, LMP-2A/B, and BNRF1) in malignant cells. Each dot represents a single cell, and the depth of colour from grey to red represents low to high expression. **d** Violin plots showed the normalised expression of cluster markers, chemokines, and genes associated with NF-κB and Notch pathways in each cluster. In each plot, cell clusters and the expression level of a gene as the chart tile are indicated at the x- and y-axis, respectively. **e** Representative images of multiplex immunofluorescence staining of malignant cells in NPC tissues. Proteins detected using respective antibodies in the assays are indicated on top. The red, green, and orange arrows indicated the representative cells positive for EPCAM, LMP1, and co-expression of EPCAM and LMP1 proteins in malignant cells, respectively. Images are representative of three biological replicates. Scale bars, 50 μm. **f** Bar plots showed the selected signalling pathways with significant enrichment of GO (top panel) and KEGG (bottom panel) terms for EBV$^+$ malignant cells (EP_C1_LMP1) compared to EBV$^-$ malignant cells (EP_C2_EPCAM), coloured from light to dark according to their $-\log_{10}$(P-values) from low to high. P-values were calculated by one-sided hypergeometric test and adjusted for multiple comparisons.

tumour samples (Supplementary Fig. 10f). Noteworthily, the P02, P08, P11, and P15 samples with a higher proportion of C4 cluster compared to other samples showed enrichment of cell cycle (E2F, MYC, and G2M checkpoint) related pathways (Supplementary Fig. 10e–g), although C4 cluster had the highest proliferation scores but overall low content in NPC tumours compared to the other three clusters. These observations suggest the intra- and inter-tumour heterogeneity of the malignant cells in NPC.

**Intercellular interaction network in NPC.** To explore the cellular communication network in NPC, we examined potential ligand-receptor binding among different cell clusters derived from NPC tumours and PBMC, using CellPhoneDB software[35] (Supplementary Data 6). We observed intensive cellular interactions among the DC_C3_LAMP3 cells, Treg cells, and exhausted CD8$^+$ T cells (CD8_C11_PDCD1) via inhibitory, co-stimulatory molecules, or chemokines (Fig. 6a, b). Among them, DC_C3_LAMP3 cells were predicted to interact with Treg_C1_SELL cells in peripheral blood through CCL17-CCR4 and CCL22-CCR4, which are known for recruiting Treg cells into tumour tissue[36] (Fig. 6a). Treg_C4_TNFRSF4 cells had high expression of CTLA4, ENTPD1, and CSF1, which showed ligand-receptor bindings to CD80/CD86, ADORA2A, and SIRPA on DC_C3_LAMP3 cells, suggesting the potential interaction between Treg_C4_TNFRSF4 and DC_C3_LAMP3 cells (Fig. 6a). DC_C3_LAMP3 cells were also predicted to interact with CD8_C11_PDCD1 cells through CD200-CD200R signalling, a non-classical immune-suppressive pathway involved in the suppression of anti-tumour responses[37] (Fig. 6b). Potential ligand-receptor interactions were observed between Treg_C4_TNFRSF4 and CD8_C11_PDCD1 cells, including those of chemokines (CCL4-CCR8), adhesive connection (ITGAL-ICAM1 and SELPLG-SELL), and immune regulation (HAVCR2-LGALS9; Fig. 6b and Supplementary Data 6), which are well-known in the TME of tumour and promote the immune-suppressive activity of Treg cells and CD8$^+$ T cells exhaustion[26,38]. Notably, these potential interactions were commonly observed in our NPC cohort (Supplementary Fig. 11a). Consistently, in another independent NPC sample collection (n = 113), we observed strong correlations of expression among the gene signatures for DC_C3_LAMP3 cells, Treg cells, and exhausted CD8$^+$ T cells (CD8_C11_PDCD1; r > 0.8, P < 2.2 × 10$^{-16}$; Supplementary Fig. 11b). These observations suggest the widespread occurrence of the immune-regulatory interactions among DC_C3_LAMP3, Treg_C4_TNFRSF4, and CD8_C11_PDCD1 cells in NPC tumours. We further performed multiplex immunohistochemistry (IHC) staining of NPC biopsies and confirmed the physical juxtapositions of CD80-expressing DC_C3_LAMP3 cells (CD80$^+$) and CTLA4-expressing Treg cells (CD3$^+$CD4$^+$FOXP3$^+$), as well as PD-L1-expressing DC_C3_LAMP3 cells (CD80$^+$) and PD-1-expressing CD8$^+$ T cells (CD3$^+$CD8$^+$; Fig. 6c, d).

Between malignant NPC cells and immune cells, we observed that EBV$^+$ EP_C1_LMP1 cells had significantly more receptor-ligand interactions than EBV$^-$EP_C2_EPCAM cells in each NPC patient (Supplementary Fig. 12a). We noted that EP_C1_LMP1-cells uniquely expressed CX3CL1 in tumour, which was predicted to interact with CX3CR1 expressed on peripheral immune cells including CD8_C5_CX3CR1 cells, DC_C1_FCER1A cells, NK cells, and monocytes (Supplementary Fig. 12b), suggesting the chemotactic potential of EP_C1_LMP1 cells to immune cells from peripheral blood. Moreover, we observed that EGFR on EP_C1_LMP1 cells was predicated to bind TGFB1 on multiple cell types, which has been reported to regulate the EBV life cycle[39] (Supplementary Fig. 12c). We also observed potential interacting pairs between EP_C1_LMP1 cells with activated Notch pathway and multiple cell types through NOTCH1-TNF and NOTCH2-JAG2, which have been related to radiation sensitivity[40] and cancer stem-like side population cells[41] in NPC (Supplementary Fig. 12d). In addition, the above-mentioned interactions were commonly determined but with individual variable intensity among the patients (Supplementary Data 6), suggesting that the interactions are widespread phenomena and heterogeneous in NPC (Supplementary Fig. 12b–d).

## Discussion

Through the comprehensive single-cell transcriptome study on NPC, we provided a landscape view of the heterogeneous cell composition and complex interacting network in the tumour microenvironment and peripheral circulating blood of NPC at single-cell resolution. Transcriptome analyses of more than 176,000 individual cells of 53 subtypes revealed two distinct microenvironments between tumour and peripheral blood in NPC. With such large-scale single-cell data, we identified novel cell populations with specific gene signatures in NPC. Moreover, in combination of TCR repertoire sequencing, we delineated the potential developmental trajectories of intratumoral immune cells. Furthermore, we dissected a multiple intercellular network in NPC using ligand-receptor paring analyses (Fig. 7).

CD8$^+$ T cells are the key effector of anti-tumour immunity with cytotoxicity to kill tumour cells[42]. We observed an abundance of tumour infiltrating CD8$^+$ T cells in NPC, which exhibited clonal expansion, effector, proliferation, and exhausted status, suggesting that the CD8$^+$ T cells were largely suppressed amid being stimulated by the tumour neoantigens in the TME of NPC. This observation is consistent with the previous findings in other cancer types[11,43], suggesting a common immunosuppressed state of CD8$^+$ T cells in tumours. Combining TCR repertoire and transcriptome analyses for each T cell, we revealed, for the first time to our knowledge, the differentiation trajectory of CD8$^+$ T cells in NPC, by which CX3CR1$^+$CD8$^+$ T cells

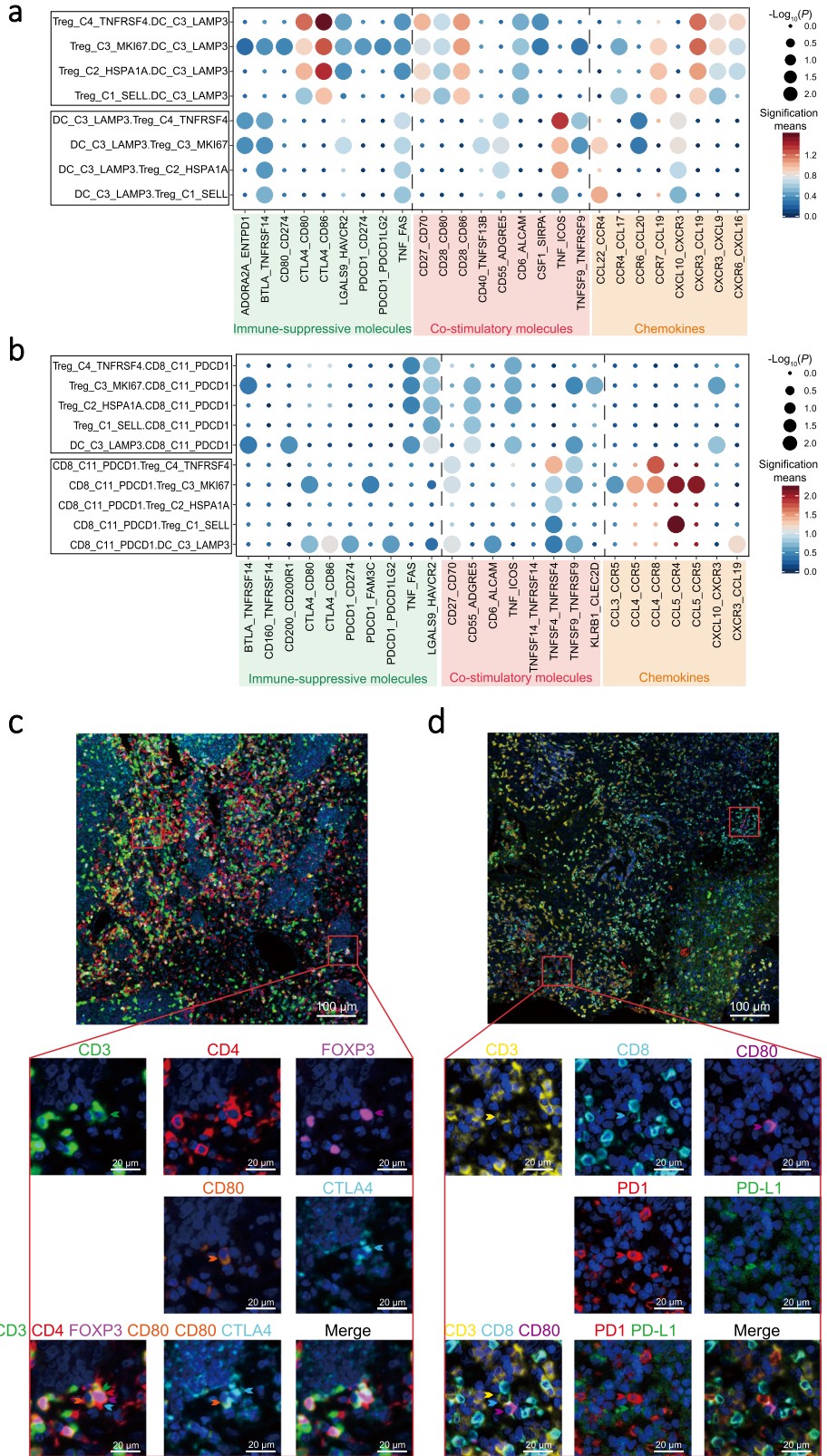

(CD8_C5_CX3CR1) in the peripheral blood infiltrated and transformed to the exhausted CD8$^+$ T cells (CD8_C11_PDCD1) in tumours. Consistently, the CX3CR1$^+$CD8$^+$ T cells of highly cytotoxic potential shared TCR clonotypes with tumour infiltrating CD8$^+$ T cells, which are responsible for recognising antigens. In light of the previous findings that CX3CR1$^+$CD8$^+$ T cells are essential for viral control and infiltrating to tumour site

to reduce tumour growth[44,45], our findings raise possible T cell therapies for NPC by infusing the CX3CR1$^+$CD8$^+$ T cells from peripheral blood after ex vivo expansion or engaging chimeric TCR of T cells with CX3CR1 chemotactic potential towards tumour site.

As cancer cells grow with high immunogenicity, inflammatory cells that are actively immunosuppressive, including regulatory

**Fig. 6 Intercellular interactions among immune and malignant cells in NPC. a, b** Dot plots showed selected ligand-receptor interactions between two cell clusters, for Treg and DC_C3_LAMP3 cells (**a**) and for exhausted CD8[+] T (CD8_C11_PDCD1) and DC_C3_LAMP3 cells (**b**). The ligand-receptor interactions and cell-cell interactions are indicated at columns and rows, respectively. The means of the average expression levels of two interacting molecules are indicated by colour heatmap (right panel), with blue to red representing low to high expression. The $\log_{10}$(P-values) were indicated by circle size in one-sided permutation test. Different colour boxes at the bottom represent different function modules of receptor-ligand interactions. **c** Representative images of multiplex IHC staining for the juxtaposition of CTLA4-expressing Treg cells (CD3[+]CD4[+]FOXP3[+]) and CD80-expressing DC_C3_LAMP3 cells in NPC tissue samples. Proteins detected using respective antibodies are indicated on top. The green, red, magenta, cyan, and orange arrows indicated positive cells with the expression of CD3, CD4, FOXP3, CTLA4, and CD80 proteins in NPC tissue, respectively (bottom panel). Images are representative of three biological replicates. Scale bars, 100 μm and 20 μm for top and bottom panels, respectively. **d** Representative images of multiplex IHC staining for the juxtaposition of PD1-expressing CD8[+] T cells (CD3[+]CD8[+]) and PD-L1-expressing DC_C3_LAMP3 cells (CD80[+]) in NPC tissue samples. Proteins detected using respective antibodies are indicated on top. The yellow, cyan, magenta, red, and green arrows indicated positive cells with the expression of CD3, CD8, CD80, PD1, and PD-L1 proteins in NPC tissue, respectively (bottom panel). Images are representative of three biological replicates. Scale bars, 100 and 20 μm for top and bottom panels, respectively.

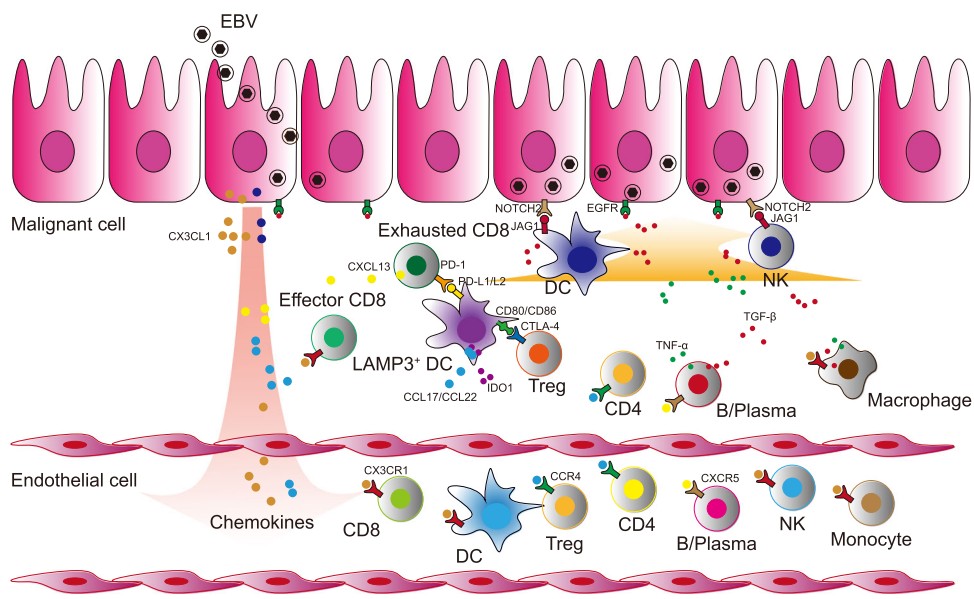

**Fig. 7 Schematic diagram of cross-talks among multiple immune cells in the TME of NPC.** EBV infects nasopharyngeal epithelial cells and participates in the tumorigenic process of NPC. EBV-positive malignant NPC cells secret a variety of chemokines (CX3CL1, etc.) and initiate the recruitment and tumoral infiltration of multiple immune cells with the chemokines receptors from the peripheral blood. Multiple tumour infiltrating immune cells activate EGFR and Notch pathway in EBV-positive malignant NPC cells. Naive CD8[+] cells infiltrate to the lesion and develop to effector and further exhausted CD8[+] cells. Peripheral DCs infiltrate to the tumour and differentiate into LAMP3[+] DCs. The mature LAMP3[+] DCs with the expression of PD-L1/PD-L2 interact with PD1 on CD8[+] T cells whereby the signalling restrains the activation of CD8[+] T cells and promotes their exhaustion. Treg cells interact with LAMP3[+] DCs through CTLA4-CD80/CD86, which might limit the antigen presentation process of DCs and promote the secretion of IDO1 to induce the proliferation of Treg cells. The intensive cell-cell interactions among LAMP3[+] DCs, Treg cells, exhausted CD8[+] T cells, and malignant cells foster an immune-suppressive niche for the tumour microenvironment of NPC.

T cells are recruited to help evade immune destruction by suppressing cytotoxic lymphocytes[46]. We observed three heterogeneous Treg cell clusters in NPC tumours, each with specific gene signatures and functions. Treg_C4_TNFRSF4 cells featured with high expression of TNFRSF family genes (*TNFRSF4*, *TNFRSF9*, and *TNFRSF18*) as well as *CCR8* and exhibited the strongest immune-suppressive function compared to other Treg cell clusters in NPC. Consistently, it has been reported previously that Treg cells with the expression of TNFRSF family genes facilitate tumour immune evasion and promote cancer development[36,47]. Moreover, high expression of *CCR8* has been demonstrated as a signature of Treg cells that restrain immunity, of which their amount in tumours is significantly associated with poor prognosis in several cancers[26,48]. Indeed, our present study revealed that the fraction of Treg_C4_TNFRSF4 in Treg cells might contribute to the poor survival of NPC. Given that CCR8 can promote immune-suppressive Treg cells[26], a blockade of CCR8 signalling in Treg cells might abolish their specific

suppressive effect on cytotoxic lymphocytes and thus inhibit tumour growth. While the presence of tumoral Treg cells in NPC has been confirmed using IHC staining assays in our present study and previously by other group[15], the origin of tumour infiltrating Treg cells in NPC remains exclusive. Pseudotime trajectory and TCR repertoire analyses revealed that Treg_C4_TNFRSF4 cells were differentiated from naive Treg_C1_SELL cells in peripheral blood through intermediate Treg_C2_HSPA1A or Treg_C3_MKI67 cells. We suspected that reducing the migration of Treg cells in the peripheral blood to tumour sites might attenuate the immune suppressions endowed by Treg cells. With this, we note that the high expression of *CCR4* in Treg_C1_SELL cells might be a potential therapeutic target, since it has been considered as the key molecule for Treg cells migrating into tumour[49].

Tumour-associated myeloid cells are heterogeneous as reported previously[50]. We observed multiple clusters of myeloid cells in NPC tumours, among which DC_C3_LAMP3 could be

considered as a group of regulatory and tolerogenic DC, showing high expression of the migration (*CCR7*) and maturation (*LAMP3*) related genes. Both are the signatures of LAMP3[+] DCs that have been recently demonstrated in multiple cancers[28,29]. Moreover, DC_C3_LAMP3 cells had the elevated expression of *CD274, PDCD1LG2, CD200, IDO1, EBI3, SOCS1, SOCS2,* and *SOCS3*, which are immune-suppressive related genes with similar expression pattern as LAMP3[+] DCs in lung cancer[51]. These observations suggest that DC_C3_LAMP3 cells are a group of LAMP3[+] DCs in NPC, which are a ubiquitous cell population in tumours and exert immune-regulatory function and the control of T cell activation[52]. Our data also revealed the developmental trajectory of the LAMP3[+] DCs in NPC and the master transcription factors that are potentially crucial for the promoting maturation, decreased antigen presentation capability, and increased immune regulatory capability of LAMP3[+] DCs. These transcription factors are connected to each other, forming a robust regulatory gene expression network in LAMP3[+] DCs. We suspect that targeting these transcription factors might reshape the dendritic cells towards a normal antigen presentation phenotype, leading to potential therapeutic benefits for NPC.

EBV infection is the key feature of NPC in the endemic regions, and its role in malignant transformation and tumorigenesis has been implicated in NPC[33]. We identified two groups of malignant cells with (EP_C1_LMP1) or without (EP_C2_EP-CAM) EBV infection in NPC tumours. The EBV[+] NPC cells showed a distinct transcriptional state compared to the EBV[−] NPC cells, with the specific activation of major genes implicated in EBV entry and cancer-related pathways, such as NF-κB[53] and Notch pathways[54]. Moreover, the EBV[+] NPC cells had more chemotactic interactions with immune cells derived from tumours and peripheral blood, which are important for the regulation of EBV life cycle and the shaping of tumour behaviours including radiation sensitivity and stemness[40,41]. The potential chemotactic interactions might also explain the abundant infiltration of immune cells in NPC tumour stroma[55]. Given that all endemic NPC are EBV positive, our findings are of particular interests for future studies on the role of EBV infection in NPC development and the persistence of EBV in body cells compared to its gradual loss during in vitro culture. The specific presence of EBV molecules in NPC cells has been harnessed as an adoptive immunotherapy for NPC using cytotoxic T lymphocyte recognising EBV in clinical trials[56,57]. We note that the presence of TCRs recognising EBV antigen among multiple patients with NPC may engage T cells targeting NPC cells with higher specificity. However, it's noteworthy to explore whether and how the heterogeneous malignant NPC cells especially the EBV negative cells and other subtypes of cells with different proliferative capabilities contribute to the variable treatment outcomes. In addition, we didn't observe EBV transcripts in any B cells, although B cells have been known as a primary host for EBV[58]. It might be explained by the low expression level of EBV genes in infected cells and the small fraction of B cells captured in our study.

Our study revealed an intercellular network among LAMP3[+] DCs (DC_C3_LAMP3), Treg cells, and exhausted CD8[+] T cells (CD8_C11_PDCD1) in NPC, suggesting potential cross-talks among multiple immune cells to foster an immune-suppressive niche for the TME of NPC (Fig. 7). Indeed, the link between LAMP3[+] DCs and Treg cells through the interaction between CCL17-CCR4 and CCL22-CCR4 has also been demonstrated in other cancers[36], by which LAMP3[+] DCs potentiate the chemotactic recruitment of peripheral Treg cells and promote their infiltration in tumours[59]. Moreover, LAMP3[+] DCs also had abundant expression of IDO1 in NPC, which could induce the proliferation of tumour infiltrating Treg cells as reported previously[60]. On the other hand, our observation of the interaction

between tumoral Treg cells and LAMP3[+] DCs through CTLA4 and CD80/CD86 has been consistently reported in other cancers[61], whereby Treg cells regulate the maturation of tolerogenic LAMP3[+] DCs. By contrast, the cross-talks between cytotoxic CD8[+] T cells and either DCs or Treg cells have been widely addressed in multiple cancers[28,62,63]. The mutual interactions between LAMP3[+] DCs and Treg cells may enhance immune-suppressive effects on the exhausted CD8[+] T cells in NPC. Together with the observation of chemotactic potential of EBV[+] NPC cells to recruit peripheral immune cells, these findings suggest that these cross-talks among diverse cell types play important roles in maintaining the homoeostasis of TME in NPC. It would be plausible that disrupting the interactions might break the balance of TME and thus cure the tumour. Besides the promising results of PD-1/PD-L1 blockade immunotherapy in NPC, antibodies targeting EGFR have been reported to enhance the current treatment paradigms for locoregionally advanced NPC[64]. We suspect that immune-suppressive interaction of CD200-CD200R1 and LGALS9-HAVCR2 among LAMP3[+] DCs, Treg cells, and exhausted CD8[+] T cells might be also potential immunotherapeutic targets for NPC.

Taken together, through uncovering the heterogeneous tumour microenvironment of NPC at a high resolution, we identified the essential cells and molecules with potential contributions to NPC tumorigenesis, and thus provide insights into the mechanisms underlying NPC progression and the development of potential therapeutic strategies for NPC. We acknowledge that our study has several limitations. First, we observed some trends of the associations of cell compositions with clinical characteristics (Supplementary Fig. 13). Particularly, patients with more advanced NPC had a higher proportion of peripheral CD8_C10_MKI67 and more intensive cellular interactions in DC_C4_JCHAIN and NK_C2_FCER1G cells (Supplementary Fig. 13c, e). However, we are not yet to draw a conclusion with such a limited cohort size that any components of the TME could be associated with the clinical outcomes. Second, we believe that NPC is one of the malignancies that are shaped together with immunity and neoantigens endowed by the somatic alterations in malignant cells. However, we are not able to predict any neoantigens, because we have no DNA level (whole genome or exome sequencing) data with our patient samples for HLA estimations and somatic mutations callings. Lastly, although our findings of the key interaction network and molecules of the TME in NPC were obtained using robust bioinformatic analyses and additional immunostaining assays, further functional experiments are awaited to explore the biological consequences and underlying mechanisms.

## Methods

**Patient recruitment and sample collection**. Ten male individuals with nasopharyngeal carcinoma (NPC) were recruited from a local hospital in Guangzhou, China, an endemic region with high prevalence of NPC, between June 2018 and September 2018. The patients were histopathologically diagnosed with primary NPC by at least two pathologists according to the World Health Organization (WHO) classification. No history of cancer and any anti-tumour therapy prior to the primary diagnosis was self-reported. Clinical staging of NPC was determined according to the 8th edition of the International Union against Cancer (UICC) and American Joint Committee on Cancer (AJCC) staging system. Fresh tumour sample was obtained using endoscopic nasopharyngeal biopsy and matching peripheral blood sample was collected for each patient, followed immediately by single cell preparation as described below. All patients were EBV positive as confirmed using in situ hybridisation of EBV encoded small RNAs (EBERs) in tumour tissue. The average age was 50.6 and the patient's characteristics were listed in the supplementary (Supplementary Data 1). For immunostaining assays, additional NPC biopsies were collected. The specimens were collected within 30 min after the tumour resection and fixed in formalin for 48 h. Written informed consent was obtained from all participants, and the study was approved by the Institutional Review Boards at the Sun Yat-sen University Cancer Center (SYSUCC).

**Preparation of single cell suspensions**. Fresh tumour samples were processed independently with enzymatic digestion and mechanical dissociation immediately after collection to generate single cell suspensions. Briefly, each tumour was cut into small pieces with approximately 1-mm³ in a D10 resuspension buffer, containing culture medium (DMEM medium; Gibco™, USA; Cat. no. 11965092) with 10% foetal bovine serum (FBS; Gibco; cat. no. 10099141), followed by enzymatic type II (Thermo Fisher, USA; cat. no. 17101015) and IV (Thermo Fisher; cat. no. 17104019) digestion for 30 min on a rotator at 37 ℃. The digested mixture was passed through a 40 μm cell-strainer (BD Biosciences, USA; Cat. no. 352340) to obtain dissociated cells. The filtered mixture was centrifuged at 400 *g* for 5 min, and after removal of the supernatant, the pelleted cells were resuspended in 0.8% NH₄Cl red blood cell lysis buffer and incubated on ice for 10 min. After washing twice with DPBS (Gibco; cat. no. 14190250), the dissociated cells from tumour were resuspended in a sorting buffer, consisting of 1X DPBS supplemented with 0.04% BSA (Sigma-Aldrich, USA; cat. no. 9048468). Viable cells were collected using fluorescence activated cell sorting (FACS; BD FACSAria III; BD Biosciences) with negative staining of propidium iodide (PI; Thermo Fisher, cat. no. P1304MP). At least 300,000 cells were collected for each tissue sample.

From blood sample, PBMCs were isolated using a leukocyte separation solution, following the manufacture's instruction (HISTOPAQUE-1077; Sigma-Aldrich; cat. no. 10771). Briefly, 5-ml of fresh peripheral blood was collected in EDTA anticoagulant tubes (BD; Cat. no. 366643) and subsequently transferred onto the solution. After density gradient centrifugation for 20 min at 750 X g, PBMCs settled at the interphase were carefully collected and washed twice with DPBS. Residual red blood cells were lysed using the same procedure abovementioned. Viable cells were collected using FACS with PI staining.

**Library construction for single cell gene expression and TCR profiling**. Immune repertoire measurement and gene expression at single cell resolution were conducted using Chromium Single Cell V(D)J Reagent Kit (10x Genomics, USA) following the manufacturer's instructions. Briefly, the sorted cells were washed twice with the sorting buffer. Cell viability and number were determined using Trypan Blue (Thermo Fisher; Cat. no. 15250061) exclusion assay. Appropriate volume of cell suspension with a concentration of 700–1200 cells/μl were loaded in each channel, targeting a capture of 8,000 cells per sample, which were further mixed with barcoded gel beads on a Chromium Controller (10x Genomics). After reverse transcription reaction, cDNA amplification for 14 cycles was conducted on a thermal cycler (C1000; Bio-Rad, USA). The post-amplification cDNA was used as template to further enrich TCR fragments. Sequencing libraries for cDNA and TCR were separately constructed according to the instructions. The average fragment size of a library was quantitated using Qseq100 (Bioptic; Taiwan).

**Next generation sequencing and data processing**. Each DNA library was loaded into a sequencing lane on a HiSeq X system (Illumina, USA) and was sequenced with pair-end reads of 150 bp. Raw data of Binary Base Call (BCL) format was converted to FASTQ files using bcl2fastq (version v2.19.0.316, Illumina). Next, Cell Ranger pipelines (version 3.0.1; 10x Genomics) were used to align sequencing reads in the FASTQ files to reference genomes of interest and generate feature-barcode matrices. Single-cell 5'-gene expression data and TCR enriched data from the same cDNA library were processed using Cell Ranger count and Cell Ranger vdj implemented in the pipelines, respectively. The gene expression data was mapped to human genome reference sequence (GRCh38; http://cf.10Xgenomics.com/supp/cell-exp/refdata-cellranger-GRCh38-1.2.0.tar.gz) and EBV reference sequence[65] (Akata; https://github.com/flemingtonlab/public/tree/master/annotation) for cDNA sequencing reads. The TCR enriched data were mapped to the VDJ reference sequence (http://cf.10Xgenomics.com/supp/cell-vdj/refdata-cellranger-vdj-GRCh38-alts-ensembl-2.0.0.tar.gz) for TCR sequencing reads.

**Single-cell gene expression quantification and determination of cell types**. Doublets are artefactual libraries generated from two cells arising due to errors in droplet encapsulation of cells, and thus commonly affect the quality of single-cell sequencing data. The R package "DoubletFinder" (https://github.com/chris-mcginnis-ucsf/DoubletFinder) was applied to predict doublets in our data. Basically, a doublet is defined as a single-cell library representing more than one cell, and a closer examination of some known markers would suggest that the offending cluster consists of doublets of more than one cell type, while no cell type is known to strongly express both markers at the same time. We removed doublets in each sample individually, with an expected doublet rate of 0.05 and default parameters used otherwise (Supplementary Table 3). The remaining cells survived from the filtering criteria were single cells. Then the gene expression matrices for all remaining PBMC and tumour cells were combined and converted to a Seurat object using the R package Seurat (version 2.3.4, https://satijalab.org/seurat). Next, any cells were removed for which had either less than 101 UMIs, or expression of less than 501 genes, or over 15% UMIs linked to mitochondrial genes. From the remaining cells, gene expression matrices were generated with log normalisation and linear regression using the NormalizeData and ScaleData function of the Seurat package.

Because the samples were processing independently and high-dimensional variables are common in single-cell sequencing data, which might introduce potential batch effect, we used canonical correlation analysis (CCA) and RunUMAP function implemented in Seurat to reduce dimensionality and remove batch effect. Cell clusters were identified using the FindClusters function in Seurat, with a *K* parameter of 20 and default parameters used otherwise. We annotated the clusters as different major cell types based on their average gene expression of well-known markers, including CD4⁺ T cell (*PTPRC*, *CD3D*, and *CD4*), CD8⁺ T cell (*PTPRC*, *CD3D*, and *CD8A*), myeloid cell (*CD14* and *ITGAX* encoding CD11C), malignant cell (*EPCAM* and KRT family genes), B cell (*CD19* and *MS4A1*), and NK cell (*FCGR3A* and *NCAM1*).

Repeating the abovementioned steps (normalisation, dimensionality reduction, and clustering), we further identified sub-clusters and annotated them as different specific cell subtypes by the average expression of respective gene sets in each major cell type. To identify marker genes for each sub-cluster within the major cell types (CD4⁺ T, CD8⁺ T, NK, B, myeloid, and malignant cells), the expression profiles of the sub-cluster were contrasted with those of the other sub-clusters using the Seurat FindAllMarkers function. Differential expression analysis implemented in the function compared all the genes in the two datasets using the default two-sided non-parametric Wilcoxon rank sum test. A significant differentially expressed gene was determined if it had the Bonferroin-adjusted *P* value lower 0.05 and an average natural logarithm (ln) fold-change of expression of at least 0.1 and 0.25 for malignant cells and other cells, respectively. The cluster with multiple well-defined marker genes of different cell types and an elevated number of UMI was considered cell contamination and removed in downstream analysis. For each cluster (like C1) of a major cell type (like CD4⁺ T cells), we assigned a cluster identifier with a marker gene (like *LEF1*) as "CD4_C1_LEF1". The selection criteria for the marker gene included (1) with top ranking at the differential gene expression analysis for the corresponding cell cluster, (2) with strong specificity of gene expression meaning high expression ratio within the corresponding cell cluster but low in other clusters, and (3) with literature supports that it's either a marker gene or functional relevant to the type of cell.

**Collection of public single-cell datasets**. To compare the features of tumour microenvironment including cell compositions between NPC and other types of cancers, we collected the single-cell data publicly available for multiple cancers, including NSCLC[12] (downloaded from https://gbiomed.kuleuven.be/scRNAseq-NSCLC), CRC[66] (Gene Expression Omnibus; https://www.ncbi.nlm.nih.gov/geo/; GEO accession number: GSE132465), and PDAC[67] (Genome Sequence Achieve; https://bigd.big.ac.cn/gsa/; GSA accession number: CRA001160), as well as that for specific cell types, including CD4⁺ T cells in CRC[68] (GEO accession number: GSE146771), NSCLC[11] (GEO accession number: GSE99254) and HCC[28] (GEO accession number: GSE140228) and DCs in NSCLC[29] (GEO accession number: GSE127465) and HCC[28] (GEO accession number: GSE140228).

**Calculation of functional module scores**. To evaluate the potential functions of a cell cluster of interest, we calculated the scores of functional modules for the cell cluster, using the AddModuleScore function in Seurat at single cell level. The average expression levels of the corresponding cluster were subtracted by the aggregated expression of control feature sets. All analysed genes were binned based on averaged expression, and the control features were randomly selected from each bin[69]. The functional modules including proliferation score for T and malignant cells, cytotoxicity and exhausted scores for CD8⁺ T cells, IL2R, inhibitory, and co-stimulatory scores for Treg cells, as well as maturation, activation, migration, differentiation, apoptosis, antigen presentation, and immune regulatory scores for dendritic cells. The involved genes were listed in the supplementary material (Supplementary Data 3).

**Pathway enrichment analysis**. To gain functional and mechanistic insights of a cell cluster, we performed Gene Ontology (GO) and KEGG Pathway enrichment analyses using Metascape (http://metascape.org/) to identify biological pathways that were enriched in a certain gene list more than that would be expected by chance. For malignant cells, the gene list imported to Metascape included the top 100 differentially expressed genes (DEGs) with a natural logarithm of fold changes of expression (lnFC) > 0.1 in clusters. For non-malignant cells, the gene list included the top 100 DEGs with lnFC > 0.25 in clusters. *P* value <0.05 was considered to be a significant enrichment. To compare the difference of signalling pathway between two clusters (Treg_C2_HSPA1A versus Treg_C4_TNFRSF4 and DC_C2_CD1C versus DC_C3_LAMP3), we performed the gene set enrichment analysis (GSEA; version 3.0) using the selected molecular signatures database v7.0[70]. To explore the heterogeneous expression of malignant cells, we performed gene set variation analysis (GSVA, version 1.34.0), using 18 hallmark pathways described in the molecular signature database.

**Developmental trajectory inference**. To characterise the potential process of immune cell functional changes and determine the potential lineage differentiation among diverse immune cells, we performed trajectories analyses for Treg, CD8⁺ T, and dendritic cells, using Monocle2 (version 2.8.0; http://cole-trapnell-lab.github.io/monocle-release/monocle2/). The data of the indicated clusters calculated in Seurat was fed directly into Monocle2. Next, we carried out density peak clustering (Monocle2 dpFeature procedure) to order cells based on the genes with differential expression between clusters, using the differentialGeneTest function in Monocle2.

The top 1,000–2,000 significant genes (ordered by q value) were used for ordering in all instances. Then the immune cell differentiation trajectory was inferred after dimension reduction and cell ordering with the default parameters of Monocle2.

**TCR repertoire analysis**. The outputs of CellRanger vdj included the assembled nucleotide sequences for both α and β chains, the coding potential of the nucleotide sequences (that is productive or not), the translated amino acid sequence, the CDR3 sequences, and the estimated UMI value of α or β chains. Only cells with UMI values larger than 1 for α and β chains were kept. The dominant TCR of a single cell was defined based on an in-frame TCR α-β pair. If one clonotype defined as a unique in-frame TCR α-β pair was present in at least two cells, this clonotype would be considered clonal, and the number of cells with such dominant α-β pair indicated the degree of clonality of the clonotype.

R package STARTRAC (version 0.1.0) was used to assess the enrichment of TCR in various T cell clusters. The degree of clonal expansion, tissue migration, and state transition of T cell clusters upon TCR tracking were determined using three STARTRAC indices, STARTRAC-expa, STARTRAC-migr, and STARTRAC-tran, respectively. For detailed pipeline, please referred to the website (https://github.com/Japrin/STARTRAC).

**Bulk RNA sequencing and data analysis**. To identify the genes responsible for EBV infection and carcinogenesis, we compared the expression profiles between NPC and non-cancerous control cohorts. The data for the NPC cohort was retrieved from the public GEO database (Accession number: GSE102349), including 113 NPC tissue samples profiled by RNA-seq[71]. The control cohort was in-house RNA-seq data of 10 rhinitis samples that had been published recently[72]. For data analysis, pair-end reads with high quality were aligned to ribosome RNAs using Bowtie2[73], and reads after removal of those being aligned as ribosome RNAs were realigned to the human genome (GRCh38) and EBV (Akata) reference sequence using HISAT2 with default settings[74]. HTseq was used to quantitate the read counts of each gene[75]. The expression levels of genes were normalised as Transcripts Per Kilobase Million (TPM), to minimise the potential effect of tumour purity.

We next assessed the associations of the immune signatures and the survival of NPC. Receiver operating characteristic (ROC) was used to determine the optimal cut-off value of gene expression for patient stratification. Kaplan-Meier analysis was conducted to reveal the prognostic ability of normalised mRNA ratio of CCR8/FOXP3 in 88 NPC samples with prognostic information from the public cohort, and a log-rank test was performed to compare the survival between high and low normalised mRNA ratios of CCR8/FOXP3. For the correlation analyses of the expression of immune signatures genes among LAMP3+ DC (DC_C3_LAMP3), Treg cell, and exhausted CD8+ T cell (CD8_C11_PDCD1), we first selected the signature genes based on the top 200 differentially expressed genes among all cell subsets (Supplementary Data 7) and then calculated the mean of the expression (TPM) for signature genes as signature scores. Pearson correlation between signatures was calculated by cor() function in R programme.

**Assembly of context specific regulatory models and master regulator analysis of LAMP3+ dendritic cell (DC)**. A LAMP3+ DC context-specific network model of transcriptional regulation was assembled with the ARACNe[76] (https://github.com/califano-lab/ARACNe-AP), based on 357 LAMP3+ DC (DC_C3_LAMP3) single cell RNA-sequencing expression profiles. ARACNe was run with 100 bootstraps, a P value threshold of $10^{-8}$, and 0 data processing inequality (DPI) tolerance, generating a network of 38 TFs associated with 2,495 target genes by 10,759 interactions. Pearson correlation between TFs and target genes was calculated by cor() function in R. We considered the correlation coefficient between TF and downstream target genes greater than 0 as positive regulation, otherwise negative regulation.

**inferCNV analysis**. To identify malignant cells, we identified evidence for somatic alterations of large-scale chromosomal copy number variants, either gains or losses, in a single cell using inferCNV (https://github.com/broadinstitute/inferCNV), in addition to the expression of EPCAM. The raw single-cell gene expression data was extracted from the Seurat object according to the software recommendation. A public single-cell data derived from normal epithelium cells was included as a control reference[32] (GEO accession number: GSE121600). We preformed inferCNV analysis with the default parameters.

**Cellular communication analysis**. To investigate the potential cell-cell communications between any two different cell types in NPC, we performed ligand-receptor analyses using CellPhoneDB software (version 2.0.6; https://github.com/Teichlab/cellphonedb). CellPhoneDB applies an algorithm that considers only receptors and ligands with broad expression among the tested cell types, followed by calculating the likelihood of cell-type specificity of a given receptor-ligand complex with a sufficient number of permutations. The gene expression matrixes of CD8+ T, CD4+ T, NK, B, myeloid, and malignant cells were selected as input for the CellPhoneDB analysis. We identified the most relevant cell-type specific ligand-receptor interactions and considered only ligands and receptors with expression in more than 20% of the cells in the corresponding sub-clusters. Moreover, we permuted the change of cell type label for each cell at 1000 times to calculate the

significance of each pair. The P value was calculated using the proportion of the mean value for specific receptor–ligand pairs compared to a randomly permuted mean distribution. Finally, we prioritised the interactions with a P-value greater than 0.05 and selected the interaction pairs with biologically relevance.

**Immunostaining assays**. For tissue sample stored in the formalin, dehydration and embedding in paraffin were performed according to routine methods. The paraffin blocks were cut into 5 μm slides and adhered on the glass slides. The paraffin-embedded sections were dewaxed, rehydrated, subjected to the blockade of endogenous peroxidase activity, and antigen retrieval at high-temperature. Subsequently, the sections were processed further for either multiplex immuno-fluorescence (IF) or immunohistochemistry (IHC) staining assays.

Multiplex IF staining assays were conducted to determine the presence of EBV+ and EBV− NPC cells. The sections were permeabilized in PBS with 0.5% Triton X-100 (Sigma-Aldrich; Cat. no. T8787) and incubated for overnight at 4 °C with the following primary antibodies: anti-EPCAM (rabbit; Abcam, USA; Cat. no. ab71916; 1:100) and anti-LMP1 (mouse; Abcam; Cat. no. ab78113; 1:500). Subsequently, the sections were incubated with Cy3 conjugated Goat Anti-Rabbit IgG and FITC conjugated Goat Anti-Mouse IgG secondary antibodies (Servicebio, China; Cat. no. GB21303/GB22301; 1:300). Nuclei were counterstained with 4'-6'-diamidino-2-phenylindole (DAPI; Sigma-Aldrich; Cat. no. D9542). Images were captured using a confocal laser-scanning microscope (LSM880; Zeiss, Germany).

To determine the spatial contact of LAMP3+ DCs, Treg cells, and CD8+ T cells, we performed multiplex IHC staining assays using the PANO 7-plex IHC kit (Panovue, China) according to the manufacturer's instructions. The slides were incubated with blocking antibody diluent at room temperature for 10 min, and then incubated overnight at 4 °C with primary antibodies. The slides were then incubated with the secondary antibody (HRP polymer, anti-mouse/rabbit IgG) at room temperature for 10 min. Subsequently, fluorophore (tyramide signal amplification or TSA plus working solution) was applied to the sections, followed by heat-treatment with microwave. The primary antibodies were applied sequentially, followed by incubation with the secondary antibody and TSA treatment. Nuclei were stained with DAPI after all the antigens had been labelled. Multispectral images for each stained slide were captured using the Mantra System (PerkinElmer, USA). Primary antibodies included anti-CD3 (rabbit; Abcam; Cat. no. ab135372; 1:50), anti-CD4 (rabbit; Abcam; Cat. no. ab133616; 1:100), anti-CD8A (mouse; CST, USA; Cat. no. CST7030; 1:100), anti-FOXP3 (mouse; Abcam; Cat. no. ab22510; 1:100), anti-CD80 (mouse; R&D Systems, USA; Cat. no. MAB140; 1:80), anti-PD1 (mouse; CST; Cat. no. CST43248; 1:50), anti-PD-L1 (rabbit; CST; Cat. no. CST13684; 1:50), and anti-CTLA4 (rabbit; Abcam; Cat. no. ab237712; 1:100).

**Statistics analysis**. All statistical analyses were performed using R (http://www.r-project.org), including two-sided paired student t-test, two-sided Wilcoxon test, two-sided Pearson correlation test, and two-sided Kruskal–Wallis test. P < 0.05 was considered as statistical significance. Multiplex IF and IHC staining assays were confirmed in at least three biological replicates.

**Reporting summary**. Further information on research design is available in the Nature Research Reporting Summary linked to this article.

## Data availability
The raw sequence data reported in this study have been deposited in the Genome Sequence Archive (Genomics, Proteomics & Bioinformatics 2017) in the National Genomics Data Center (Nucleic Acids Res 2020), Beijing Institute of Genomics (China National Center for Bioinformation), Chinese Academy of Sciences, under accession number HRA000159 (accessible at http://bigd.big.ac.cn/gsa-human) and GEO dataset under the accession number GSE162025. The key data in this study has also been deposited in the Research Data Deposit (RDDB2020000980; http://www.researchdata.org.cn/). Other datasets described in the Methods could be downloaded from NCBI GEO under the accession numbers GSE132465[66], GSE146771[68], GSE99254[11], GSE140228[28], GSE127465[29], GSE102349[71], and GSE121600[32], Genome Sequence Achieve under the accession numbers CRA001160[67], and the URL https://gbiomed.kuleuven.be/scRNAseq-NSCLC[12]. The remaining data are available within the Article, Supplementary Information or available from the authors upon request.

## Code availability
The scripts are available at https://github.com/bei-lab/scRNA-of-NPC[77].

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

## Acknowledgements

This study is supported by the national Key research and development program of China (2016YFC0902001), the Sci-Tech Project Foundation of Guangzhou City (201707020039), Guangdong Innovative and Entrepreneurial Research Team Program (2016ZT06S638), the National High Technology Research and Development Program of China (2012AA02A206), the National Program for Support of Top-Notch Young Professionals (J.-X.B.), Chang Jiang Scholars Program (J.-X.B.), Special Support Program of Guangdong (J.-X.B.) and Sun Yat-sen University Young Teacher Key Cultivate Project (17ykzd29). We thank all the participants in the study, and staffs at the High-Throughput Analysis Platform (HTAP) of SYSUCC for data generation and processing.

## Author contributions

J.X.B. and Y.X.Z. conceived and supervised this study. S.H., Y.L., W.P., Q.Y.C., J.R.C., Q.Y.W., R.J.P., P.P.W., X.G., B.L., X.J.X. and H.Q.M. performed sample preparation, library construction, and data generation. Y.L., X.L.W., S.H., B.W.H., D.M.C., G.W.L., Y.Q.L. and X.D.H. performed bioinformatics analyses under the supervision of J.X.B. and Z.Z., Y.L. and J.X.B. wrote the manuscript. All authors read and approved the final manuscript.

## Competing interests

The authors declare no competing interests.
