## [Peer Review File · Nature Communications]

Editorial note: In the second round of peer review Reviewers #1 and #2 were not available to comment on the author's response to their previous comments. Therefore, Reviewer #5 was asked to comment on the author's response to Reviewer's #1 and #2. Additionally, parts of this Peer Review File have been redacted as indicated to maintain patient confidentiality.

REVIEWER COMMENTS

Reviewer #1 (Remarks to the Author): Expert in NPC

Tumor heterogeneity and intracellular networks of nasopharyngeal carcinoma at single cell resolution

By Jin-Xin Bei, Yi-Xin Zeng and colleagues

This study examines the heterogeneous nature of the TME of NPCs which may have new insights into the complexities of the TME and responses to therapies for NPC patients. The authors used the new technology scRNA-Seq and profiled about 176 thousand cells isolated from 10 NPC tumors and matched PBMCs and paired T cell receptor-Seq. The authors performed analyses of the sequencing data obtained and examined that data for characterization of the TME and tumor infiltrating lymphocytes as well as tumor cells infected with EBV.

The analyses performed were a range of different analytics that allowed them to compare the results from matched PBMCs and TCR-Seq data. They state that they have identified a novel group of TILs LAMP3+ DCs which has elevated immunosuppressive and tolerogenic capabilities. They also state that they identified intensive interactions between the different cellular compartments which suggests cross-talk which contributes to the immunosuppressive niche for the TME in NPCs.

Transcription analyses resulted in the characterization of 53 cell subtypes broken into 4 subgroups CD8, DC, TIL, TRegs. These suggests that these cells make up the heterogenic compartments of the NPC TME and may have a role in regulating the progression and development of NPC as well as therapeutic options for patients.

My comments are below

1. The authors have done a great deal of work in sequencing these cells in an effort to understand the makeup of the TME of NPCs.
2. DC-LAMPs are not new or novel as they have been previously identified and a great deal is known about them.
3. There are statements about interactions and at best these are speculative as there are no clear evidence to support these statements.
4. The fact the TME is heterogeneous is not a new finding and what would have been useful is the comparison of how NPC may be different or unique in its composition compared to other types of cancers.
5. There are some differences between what is stated and the figures and seems to be due to a lack of editing for accuracy.
6. The vast amount of data although interesting has not been validated in any way and so the most we can tell here is that they have characterized the TME of NPC. How is it similar from what was done before with NPC RNA-Seq?
7. In terms of intracellular interactions what are the key areas that clearly spell out these findings as one cannot make these leaps with validation what is found by using standard acceptable functional assays as well as technologies that can also show similar patterns.
8. Please identify 2 areas that have important meaning and then validate and check functionally for some of the markers as well.
9. It would be helpful for the authors to try and assimilate the large body of data already existing instead of just new data that has no mechanistic support.
10. Placing such a large amount of data without any validation will just create more confusion in the

field and the authors should make every effort to address this issue.

Reviewer #2 (Remarks to the Author): Expert in scRNA-seq

This paper reports a comprehensive catalog of single-cell RNA expression phenotypes across 176,447 cells from 10 NPC tumors. The authors collate, correlate and filter the data with a variety of top of class algorithms to organize the data and derive various putative inferences. Among the many novel observations, the authors identify a subset of LAMP3+ dendritic cells that potentially exist in a (virtual) niche with Treg and exhausted CD8+ cells.

The study is comprehensive and unique. It is at this stage observational, and no experimental validation of any of the suppositions was performed. However, that should NOT delay publication of the manuscript as the effort will inform the work of many others. I would not recommend any follow up wet-lab experiments for this manuscript.

The authors do not overstate their findings and keep the conclusions as postulates. I would however recommend that should the paper be accepted, the manuscript is carefully edited to keep the conclusions as "correlations" pending experimental validation.

The figures are clear, the suppositions and conclusions they make are reasonable. Several of the algorithms are just listed as obscure names, like DEG. The paper would benefit when each new algorithm is introduced to give a brief explanation of the technique, and not just refer to the original paper. This will help others reading the paper to better follow the context of the conclusions.

I was curious, however, about why the authors did not search in the carcinoma cells (I presume they were in the dataset) for neo-antigens? Or is NPC not subject to neoantigen development? In either case, it should be noted.

Second, although the cohort size was modest, was there any clinical outcomes to which the expression profiles could be correlated? I understand the authors don't want to call anything a "diagnostic" as that often invites criticism, but it would be good in the discussion to either mention whether there are any tentative conclusions worth following up in a larger cohort.

This report will make a unique and important addition to the growing collection of human tumor/immune single cell datasets.

Reviewer #3 (Remarks to the Author): Expert in immuno-oncology

In this interesting manuscript, " Tumour Heterogeneity and Intercellular Networks of Nasopharyngeal Carcinoma at Single Cell Resolution", Dr. Yang Liu et al. revealed the landscape of tumor and infiltrating immune cells in NPC. The study identified 176,447 cells from 10 NPC tumors and matched peripheral blood, using scRNA-seq and TCR-seq, delineated the immune repertoire of

NPC. I would like to congratulate the authors for detailed analyses of the tremendous amount of data. The data generated is interesting and could be a good addition to the field. However, the small sample size and previous similar publications have decreased its novelty.

Major comments:

1. A recent paper on single cell profiling of tumor cells and infiltrating immune cells of NPC published in 2020 by Jin Zhao et al. in Cancer Letters, which decreases the novelty of the current work. So the statement of “first study” is not appropriate. I don't think the authors cited and discussed the similar and novel findings compared to that study. For example, in the study Jin Zhao et al, NPC exhibited a high extent of intratumor heterogeneity, and malignant cell separated clearly in the tSNE map, which seems very different from the current data.
2. The authors spent most of efforts on the “intra-tumor heterogeneity”. What about the inter-patient heterogeneity? Did the TME components, NPC cells, interaction between subsets of immune, between immune cells and NPC cells similar or different in each patient? Is there any trend that age, gender, stage etc. impacts these relationships?

Following are some minor comments and advices.

1. The patient information should be clearly presented since there are only 10 patients. Patient's characteristics may be important to interpret the data.
2. Although NPC is was EBV tested in these patients either by antibody or PCR
3. Most patients had stage III-IV disease. Was the tissues from FNA or core biopsy?
4. How to distinguish doublet and single cells in supplementary figure 1B?
5. The study identified 176,447 cells. All cells types in TME and blood should be included, however, there were no fibroblasts and endothelial cells that are commonly found in TME of other cancer types. What was the reason?
6. Line 110, what are the 'epithelia cells'? Malignant cells? Non-malignant cells?
7. What are the criteria to select the 'gene' at the end of different T/B cell cluster names.
8. Line 221, please clarify how to calculate the IL2R scores in 'Method' section.
9. It is not convincing to define EP_C1_LMP1 as NPC tumor cells, provide more approaches such as inferred CNVs.
10. Suggest plotting the epithelial cells by patients in addition to Figure 5A.
11. Describe intertumoral heterogeneity and intratumoral heterogeneity of malignant cells of NPC.
12. Line 567-568, please mind the unidentified characteristics.

Reviewer #4 (Remarks to the Author):Expert in immuno-oncology

The authors present single cell RNA data for nasopharyngeal carcinoma for tumor and peripheral blood. While this data is an important road map to understand cell signatures with this type of cancer, the data are highly descriptive with too many suggestive correlations that are not backed up with sound science. In particular, there is only one sub-figure that shows protein expression to back up the RNA data. With the paucity of validating protein data it is hard to interpret most of the claims in this manuscript. Hence, most of what is claimed is suggestive at best. There are also techniques/signatures that are not explained very well so it was extremely hard to review some of

the data as it was not clear what the authors' were claiming. In summary, if the authors resubmit this manuscript there either has to be protein data to back up their claims or the manuscript needs to be rewritten to take out all of the unsubstantiated claims throughout the results and discussion sections.

Point-by-point responses to reviewers' comments

Reviewer #1 (Remarks to the Author): Expert in NPC

1. The authors have done a great deal of work in sequencing these cells in an effort to understand the makeup of the TME of NPCs.

Response: We are very grateful to the Reviewer #1 for the favourable comments on our study as well as the following suggestions that are helpful to improve our manuscript.

2. DC-LAMPs are not new or novel as they have been previously identified and a great deal is known about them.

Response: Thanks for the comment. We agree with the Reviewer #1 that Lysosome Associated Membrane Proteins (LAMPs) have been involved in many different aspects of cell biology and cancer progression (Alessandrini et al., *Semin Oncol* 2017, 44(4):239-253). The LAMP family has five members so far, including LAMP1 (CD107a), LAMP2 (CD107b), LAMP3 (DC-LAMP), LAMP4 (Macrosialin or CD68) and LAMP5 (BAD-LAMP). LAMP1 and LAMP2 are common proteins in the lysosomal membrane and ubiquitously expressed in human cells. By contrast, LAMP3 and the other two are cell-type specific proteins (Wilke et al., *BMC Biol* 2012, 10:62; Defays et al., *Blood* 2011, 118(3):609-17). Consistently, we observed in our dataset that LAMP1 and LAMP2 were widely expressed in almost all myeloid cells; LAMP3 was considered as the marker of mature DC (LAMP3⁺ DC); macrophages had high expression level of CD68 (LAMP4), and LAMP5 was specifically expressed in plasmacytoid dendritic cells (pDCs; **Response Figure 1A**).

DC-LAMP or LAMP3 precisely has been a hotspot in cancer research most recently, largely due to the accumulating evidence of the involvement of LAMP3⁺ DCs in TME as revealed by single-cell RNA sequencing. For example, LAMP3⁺ DCs highly expressed CD274 (PD-L1) and PDCD1LG2 (PD-L2) in hepatocellular carcinoma (HCC; Zhang et al., *Cell* 2019, 829-845.e20) and non-small-cell lung cancer (NSCLC; Maier et al., *Nature* 2020, 580(7802):257-262), which were predicted to bind to PDCD1 (PD1) on multiple T cells, suggesting their potential role in regulating T cells in TME.

No publication results were found in PubMed using the keywords “nasopharyngeal carcinoma” and “LAMP3 or DC-LAMP” as of Sep 4, 2020. For the first time, to the best of our knowledge, our study revealed the presence and potential contribution of LAMP3+ DCs (DC_C3_LAMP3) in the TME of NPC. First, we identified LAMP3+ DC as a unique regulatory and tolerogenic group of DC in NPC, with high expression of additional genes including *CD274* (PD-L1), *PDCD1LG2* (PD-L2), and *CD200* (manuscript **Supplementary Figure 9a**, reproduced below as **Response Figure 1B**). Trajectory analysis revealed that LAMP3+ DCs migrated from peripheral blood into the tumour site and turned to maturation state (manuscript **Figure 4d** and **4e**), which is consistent with that in HCC (Zhang et al., *Cell* 2019, 829-845.e20). Moreover, LAMP3+ DCs shared similar expression pattern of immune-suppressive related genes among multiple cancers (Zilionis et al., *Immunity* 2019, 50(5):1317-1334.e10; Zhang et al., *Cell* 2019, 829-845.e20; manuscript **Supplementary Figure 9a**, reproduced below as **Response Figure 1B**). We further identified the decreased antigen presentation ability and increased immune regulation of LAMP3+ DCs in NPC. Second, we identified transcription factors (TFs) related to the maturation, decreased antigen presentation ability, and immune-regulatory ability of LAMP3+ DCs (manuscript **Figure 4f** and **Supplementary Figure 9c**). Third, we performed cell-cell interaction analysis, which revealed that LAMP3+ DCs potentially interacted with exhausted CD8+ T cells in NPC tumour through the receptor-ligand pairs of CD274-PDCD1 (PD-L1-PD1) and CD200-CD200R, and also with Treg cells through CTLA4-CD80 pair (manuscript **Figure 6b**; previous **Supplementary Figure 8C**). More importantly, we provided additional evidence to validate the interactions among the abovementioned cell types, using multiplex immunohistochemistry (IHC) staining assays in this revision. We observed the co-localization of CTLA4 and CD80 on Treg cells and LAMP3+ DCs, as well as PD-L1 and PD1 on LAMP3+ DCs and CD8+ T cells, respectively (manuscript **Figure 6c** and **6d**, reproduced below as **Response Figure 2A** and **2B**), suggesting spatial feasibility for their cell-cell interactions in NPC (please kindly check our response to the Reviewer #1’s point 3 below, Pages 4-8).

Taken together, our study identified, for the first time to the best of our knowledge, LAMP3+ DCs in NPC tumour and their cross-talks with Treg cells and exhausted CD8+ T cells, which might jointly foster an immune-suppressive niche for the TME of NPC (manuscript

Figure 7; previous **Supplementary Figure 9**). These findings would provide insights into the development of therapeutic strategies of NPC by targeting the key molecules in the cross-talks and TME.

In this revision, we have added the new results as mentioned above. Please kindly check at the section “Intercellular interaction network in NPC” in the **Results**, the 2nd paragraph, Page 16 to the 1st paragraph, Page 17.

Response Figure 1. Gene expression profile of myeloid cells.

A. Violin plots showed the expression levels of LAMP family (LAMP1, LAMP2, LAMP3, CD68, and LAMP5) in myeloid cells derived from NPC tumours. Cell clusters were indicated at the

x-axis, and the normalization expression level was indicated at y-axis for each gene as a row panel.

B. Heatmap showed the normalized mean expression of cluster markers and immune-suppressive genes (rows) in dendritic cell clusters (columns) derived from NPC (green in Source), NSCLC (non-small-cell lung cancer; blue in Source), and HCC (hepatocellular carcinoma; red in Source). Filled colours from purple to yellow represent scaled expression levels from low to high, respectively.

3. There are statements about interactions and at best these are speculative as there are no clear evidence to support these statements.

Response: Thanks. In our study, potential cell-cell interactions were inferred using CellPhoneDB software (Vento-Tormo et al., *Nature* 2018, 563(7731):347-353), which revealed promising immune-suppressive (CD274-PDCD1 or PD-L1-PD1, CD80-CTLA4) and chemokine (CCL17-CCR4, CCL4-CCR8) interactions between LAMP3+ DCs, Treg cells, and exhausted CD8⁺ T cells in NPC. In this revision, we performed additional multiplex IHC staining assays and observed the co-localization of CD80-expressing LAMP3+ DCs and CTLA4-expressing Treg cells, as well as PD-L1-expressing LAMP3+ DCs and PD1-expressing CD8+ T cells (manuscript **Figure 6c** and **6d**, reproduced below as **Response Figure 2A** and **2B**), indicating the spatial contacts between LAMP3+ DCs and Treg cells through CD80-CTLA4, as well as between LAMP3+ DCs and CD8+ T cells through CD274-PDCD1 (PD-L1-PD1) in NPC. These observations are consistent with their ligand-receptor interactions predicted using CellPhoneDB, suggesting the robustness of the analytic pipeline.

Indeed, CellPhoneDB had been utilised in many studies to infer cell-cell interaction networks, which were also verified by functional experiments (Chua et al., *Nat Biotechnol* 2020, 38(8):970-979; Lee et al., *Nat Genet* 2020, 52(6):594-603; Zhang et al., *J Hepatol* 2020, S0168-8278(20)30360-3). In our study, significant ligand-receptor interactions were identified with common occurrence in most patients (manuscript **Supplementary Figure 11a** and **12b-d**, reproduced below as **Response Figure 3**). These studies again suggest the robustness of the software and analytic pipeline. CellPhoneDB applies an algorithm that considers only receptors and ligands with broad expression among the tested cell types,

followed by calculating the likelihood of cell-type specificity of a given receptor-ligand complex with a sufficient number of permutations. Moreover, CellPhoneDB further prioritizes interactions that are highly enriched between cell types based on the number of significant receptor-ligand pairs and manually selected based on biological relevance.

In this revision, we have updated the above data in the **Results** section. Please kindly check at the section “Intercellular interaction network in NPC”, the 2nd paragraph, Page 16 to the 1st paragraph, Page 17. We have added a brief introduction of CellPhoneDB in the **Methods** section. Please kindly check at the section “Cellular communication analysis”, the 2nd paragraph, Page 31. We have also acknowledged that “further functional experiments are awaited to explore the biological consequences and underlying mechanisms.” Please kindly check at the **Discussion** section, the 2nd paragraph, Page 21 to the 1st paragraph, Page 22.

Response Figure 2. Representative images of multiplex immunohistochemistry staining (IHC) in NPC samples

A. Representative images of multiplex IHC staining in NPC sample for the juxtaposition of CTLA4-expressing Treg cells (CD3+CD4+FOXP3+) and CD80-expressing LAMP3+ DCs (CD80+; top panel; Scale bars, 100µm). Proteins used in the assays are indicated on top. The green, red, magenta, cyan, and orange arrows indicated the expression of CD3, CD4, FOXP3, CTLA4, and CD80 proteins in NPC tissue, respectively (bottom panel; Scale bars, 20µm).

B. Representative images of multiplex IHC staining in NPC sample for the juxtaposition of PD1-expressing CD8+ T cells (CD3+CD8+) and PD-L1-expressing LAMP3+ DCs (CD80+; top panel; Scale bars, 100µm). Proteins used in the assays are indicated on top. The yellow, cyan, magenta, red, and green arrows indicated the expression of CD3, CD8, CD80, PD1, and PD-L1 proteins in NPC tissue, respectively (bottom panel; Scale bars, 20µm).

Response Figure 3. Inter-patient heterogeneity of cell-cell interactions in NPC.

A. Dot plot showed selected ligand-receptor interactions between DC_C3_LAMP3 and Treg_C4_TNFRSF4 or CD8_C11_PDCD1 (rows) in each patient (columns). Dot plots showed the interactions of CX3CL1-CX3CR1 (**B**), TNF-NOTCH1 and JAG1-NOTCH2 (**C**), and EGFR-TGFB1 (**D**) between EBV+ malignant cells (EP_C1_LMP1) and other immune cells (rows) in each patient (columns). P values were indicated by circle size (scale at the right;

permutation test). The means of the average expression levels of two interacting molecules were indicated by colour heatmap (right panel). Patient ID was indicated at the bottom of each plot.

4. The fact the TME is heterogeneous is not a new finding and what would have been useful is the comparison of how NPC may be different or unique in its composition compared to other types of cancers.

Response: Thanks. We agree with the Reviewer #1 that the heterogeneous nature of TME is common in malignancies. We want to note that our present study delineated the most comprehensive profile of cell composition in NPC, with 53 cell types and the largest number of more 176,447 cells by far. With such large-scale of data, we identified the diverse phenotypes of immune cell in NPC and projected the trajectory of exhausted CD8+ T and immune-suppressive TNFRSF4+ Treg cells in tumours derived from CX3CR1+CD8+ T and naive Treg cells in peripheral blood, respectively (manuscript **Figure 2c** and **3c**). We also explored the heterogeneity of malignant cells, especially with the status of EBV infection (manuscript **Figure 5b** and **5c**). Moreover, we predicted and validated the immune-suppressive interactions (CD274-PDCD1 or PD-L1-PD1, CTLA4-CD80) among LAMP3+ DCs (DC_C3_LAMP3), Treg cells, and exhausted CD8+ T cells (CD8_C11_PDCD1; manuscript **Figure 6a** and **6b**, previous **Figure 6C** and **Supplementary Figure 8C**; manuscript **Figure 6c** and **6d**, reproduced as **Response Figure 2A** and **2B**). Together, our findings dissected the heterogeneous TME of NPC in greater details and identified key molecules and interacting networks among them at single-cell resolution, which provided insights into understanding the important role of maintaining the immune-suppressive niche in NPC progression and developing potential therapeutic strategies by disrupting the cell-cell network in the TME of NPC.

As suggested, we compared cell compositions between NPC and other types of cancers with single-cell data publicly available, including non-small-cell lung cancer (NSCLC; Lambrechts et al., *Nat Med* 2018, 24(8):1277-1289), colorectal cancer (CRC; Lee et al., *Nat Genet* 2020, 52(6):594-603), and pancreatic ductal adenocarcinoma (PDAC; Peng et al., *Cell Res* 2019, 29(9):725-738). First, individual heterogeneity of cell composition was observed in

each type of cancer and infiltrating immune cells including T, B, and myeloid cells were common in all types of cancer (manuscript **Supplementary Figure 1g**, reproduced below as **Response Figure 4A**). Second, we observed a significantly higher proportion of T cells in NPC compared to any other cancers (manuscript **Supplementary Figure 1h**, reproduced below as **Response Figure 4B**). The abundance of infiltrating lymphocytes containing mainly prominent T cells was consistent with previous findings that tumour infiltrating leucocytes was a major characteristic of NPC tumour stroma (Perry et al., *Immunol Today* 1998, 19(9):414-21). It has been demonstrated that infiltrating immune cells could secrete cytokines (like TGF- β and CCL20) to support abnormal growth of epithelial cells and tumorigenesis in NPC (Gourzones et al., *Semin Cancer Biol* 2012, 22(2):127-36). Third, tissue-specific cells could be captured, such as alveolar cells in NSCLC and acinar cells in PDAC (manuscript **Supplementary Figure 1g**, reproduced below as **Response Figure 4A**).

In this revision, we have added the comparisons in the main text, as well as the figures in the supplementary material. Please kindly check at the section ““Landscape view of cell composition in tumour biopsy and PBMC in patients with NPC” in the **Results**, the 1st paragraph, Page 5.

Response Figure 4. Comparison of heterogeneous cell composition among four types of cancers.

A. Bar plots showed the proportion of each cell type in NPC, NSCLC (non-small-cell lung cancer), CRC (colorectal cancer), and PDAC (pancreatic ductal adenocarcinoma). Each bar

represents an individual patient, with cell proportions coded as different colours for cell types on the right.

B. Box plots showed the proportion of T cell in NPC, NSCLC, CRC, and PDAC. Each comparison was tested by using Two-sided Wilcoxon test.

5. There are some differences between what is stated and the figures and seems to be due to a lack of editing for accuracy.

Response: Thanks. We apologize for the overlooked mistakes. In this revision, we have corrected these, including that 1) the row name of *CCL20* in manuscript **Figure 4b** in our previous submission was corrected as *CCL22*; and 2) we unified *ITGAE* gene in the supplementary figure and its alias *CD103* in text (manuscript **Supplementary Figure 6a**; previous **Supplementary Figure 4C**) as *ITGAE*.

6. The vast amount of data although interesting has not been validated in any way and so the most we can tell here is that they have characterized the TME of NPC. How is it similar from what was done before with NPC RNA-Seq?

Response: Thanks for the comment. In this study, we dissected the TME of NPC with comprehensive details, using tumour-blood pair design and robust analytic pipelines. Moreover, our study included new findings that were ubiquitous in our samples and further validated with additional multiplex immunohistochemistry (IHC) staining assays in this revision. Furthermore, besides the new findings, our transcriptome analysis at single-cell resolution also replicated some findings previously reported by using bulk RNA sequencing.

First, we generated the largest scale of transcriptome coupled with TCR repertoire sequencing of NPC at single-cell resolution from 10 patients, using tumour-blood pair design. These enabled us to crosscheck a finding with multiple analyses and validate findings in multiple patient samples. For example, we delineated the development trajectory of T cells based on the consistent observations across gene expression profiles, sample origin, TCR tracing, and chemokines expression. The presence of specific cell populations and interactions were consistently observed among NPC samples, reflecting the intrinsic heterogeneous nature of TME.

Second, our findings of cell populations, molecules and interactions were obtained using algorithms and analytic pipelines that had been commonly adopted by the research community, followed by prioritizations according to their functional relevance. As commented by the Reviewer #2 below, our study applied “top of class algorithms”. Specifically, we used CellPhoneDB to predict cell-cell interactions, which revealed that LAMP3+ DCs (DC_C3_LAMP3) could regulate exhausted CD8+ T cells (CD8_C11_PDCD1) via various ligand-receptor pairs (including CD274-PDCD1, PDCD1LG2-PDCD1, and CD200-CD200R pairs) and participate in the dysfunction of CD8+ T cells in NPC (manuscript **Figure 6b**; previous **Supplementary Figure 8C**). Similar findings have been demonstrated in HCC (Zhang et al., *Cell* 2019, 829-845.e20). We also observed the cellular interactions between LAMP3+ DCs and Treg cells in our single-cell dataset. In this revision, we further provided additional multiplex IHC staining and validated the presence of spatial contact of the interacting LAMP3+ DCs and Treg cells, as well as LAMP3+ DCs and CD8+ T cells (manuscript **Figure 6c** and **6d**, reproduced as **Response Figure 2A** and **2B**). Moreover, we identified the signature genes of LAMP3+ DCs (DC_C3_LAMP3), Treg cells, and exhausted CD8+ T cells (CD8_C11_PDCD1) in our single-cell dataset using DEGs (differential expression genes) analysis and we used the average expression of the signature genes (Rank Top 200) as surrogates for the proportions of each cell type in each bulk RNA-seq sample. We observed significantly strong correlation of LAMP3+ DCs and Treg cells ($r = 0.85$, $P < 2.2 \times 10^{-16}$), LAMP3+ DCs and exhausted CD8+ T cells ($r = 0.81$, $P < 2.2 \times 10^{-16}$) in an independent 113 NPC samples with bulk RNA-seq data (manuscript **Supplementary Figure 11b**; previous **Figure 6B** and **Supplementary Figure 8D**). The consistency of these results suggests the robustness of our analytic pipelines. Please also kindly refer to our response to the Reviewer #1’s point 3 above (Pages 4-8).

Third, altered expression and mutations accumulated in NF- κ B and Notch pathways have been commonly implicated in EBV infection and the development of NPC (Li et al., *Nat Commun* 2017, 8:14121; Tu et al., *Carcinogenesis* 2018, 39(12):1517-1528; Shi et al., *Int J Cancer* 2006, 119(10):2467-75). Consistently, we observed higher expression of genes involved in NF- κ B (*NFKB1*, *NFKB2*, and *REL*) and Notch (*NOTCH1*, *HES1*, and *DTX2*) pathways in the EBV+ malignant cells (EP_C1_LMP1) compared to the EBV- malignant cells

(EP_C2_EPCAM; manuscript **Figure 5b** and **5d**, reproduced below as **Response Figure 5A** and **5B**). These findings support that EBV infection and aberrant NF- κ B and Notch pathways are key events leading to NPC pathogenesis.

Moreover, a previous study demonstrated that activated NF- κ B signals could modulate the expression of multiple chemokines in EBV positive NPC cell lines, including *CXCL9*, *CXCL10*, *CX3CL1*, and *CCL20* (Chung et al., *J Pathol* 2013, 231(3):311-22). In our dataset, higher expression of *CX3CL1* and other chemokines (*CXCL3* and *CXCL8*) were also observed in EBV+ malignant cells (EP_C1_LMP1) compared to EBV- malignant cells (EP_C2_EPCAM; manuscript **Figure 5d**, reproduced below as **Response Figure 5B**; the cut-off of logFC was 0.01 and *P* value was less than 0.05 considered as statistical significance). Consistently, higher expression of *CX3CL1* was detected in another independent NPC samples (n=113) compared to non-cancerous samples (rhinitis, n=10; manuscript **Supplementary Figure 10b**; previous **Figure 5D**). With the advantage of single-cell transcriptome and tumour-blood pair design, we observed the migration of immune cells with abundant expression of CX3CR1 (the receptor of CX3CL1) from peripheral blood into tumour via the interaction of CX3CL1-CX3CR1 in NPC (manuscript **Supplementary Figure 10c**; previous **Figure 5E**). Indeed, intercellular interaction analysis revealed that EBV+ malignant cells had more interactions with immune cells compared to EBV- malignant cells (manuscript **Supplementary Figure 12a** reproduced below as **Response Figure 5C**). These findings suggest that CX3CL1 and other chemokines in EBV+ malignant cells are important molecules contributing to immune cells infiltration into tumour site.

Taken together, we provided comprehensive profiling of the TME of NPC with the largest scale of single-cell transcriptome and robust analytic pipeline. In this revision, we have compared our findings and that derived from bulk RNA-sequencing data, resulting in consistencies in the expression of genes involved in the aberrant NF- κ B and Notch pathways as well as the ligand-receptor or cell-cell interactions, which were further supported by additional IHC staining assays.

In this revision, we have added the comparison results in the **Results** section. Please kindly check at the section "Intercellular interaction network in NPC", the 2nd paragraph,

Page 16 to the 1st paragraph, Page 17; and the section “Heterogeneity of malignant cells with different EBV infection status”, the 2nd paragraph, Page 14 to the 1st paragraph, Page 15,

Response Figure 5: Heterogeneity of malignant cells in NPC.

A. UMAP plot of 2,787 epithelial cells grouped into two cell types according to their presence (EP_C1_LMP1) of EBV transcripts or not (EP_C2_EPCAM). Each dot represents a cell, coloured according to cell types.

B. Violin plots showed the normalized expression of EBV encoded molecules, cluster markers, chemokines, Notch and NF-kB pathway associated genes in each cell cluster. In each plot, cell clusters are indicated at the x-axis, and the expression level of a gene as the chart title is indicated at the y-axis.

C. Box plots showed the number of interactions of EBV+ malignant cells (EP_C1_LMP1) and EBV- malignant cells (EP_C2_EPCAM) with other cells in NPC. Each comparison was tested by using paired Student’s t-test.

7. In terms of intracellular interactions what are the key areas that clearly spell out these findings as one cannot make these leaps with validation what is found by using standard acceptable functional assays as well is technologies that can also show similar patterns.

Response: Thanks for the comment. In this study, we aimed to dissect the intrinsic cell-cell communication network in the tumour microenvironment (TME) of NPC, using intercellular interaction analysis. To better deliver our findings clearly, we have presented and interpreted the interaction results in more details in this revision.

First, intercellular interactions were inferred using software CellPhoneDB, which evaluated the interacting potential between a ligand on one cell and its receptor on another one, incorporating with their expression level, specificity, and biological functions.

Second, we observed significant interacting ligand-receptor pairs between LAMP3+ DCs and CD8+ T cells, of which CD274-PDCD1 (PD-L1-PD1) interaction is a classical immune checkpoint pathway involved in regulating TME and cancer immunotherapy (Binnewies et al., *Nat Med* 2018,24(5):541-550). Moreover, multiple interacting pairs were also observed between Treg cells and LAMP3+ DCs (DC_C3_LAMP3), including CTLA4 of Treg cells and CD80/CD86 of DC_C3_LAMP3 (manuscript **Figure 6a**; previous **Figure 6C**). It has been demonstrated that the interaction between CTLA4 expressed in Treg cells and costimulatory molecules CD80/CD86 in DCs promoted the expression of IDO1 in DCs (Fallarino et al., *Nat Immunol* 2003, 4(12):1206-12). IDO-expressing DCs have been implicated in suppressing anti-tumour T cell responses and promoting immune tolerance (Sharma et al., *Sci Adv* 2015, 1(10):e1500845). These results suggested that the Treg cells suppressed the function of LAMP3+ DCs in favour of immune tolerance to tumour cells.

Third, we performed additional multiplex IHC staining assays in this revision, which revealed the juxtaposition of CTLA4-expressing Treg cells (CD3+CD4+FOXP3+) and CD80-expressing LAMP3+ DCs (CD80+; manuscript **Figure 6c**, reproduced as **Response Figure 2A**), as well as PD1-expressing CD8+ T cells (CD3+CD8+) and PD-L1-expressing LAMP3+ DCs (CD80+; manuscript **Figure 6d**, reproduced as **Response Figure 2B**). These observations indicate direct spatial contacts between these cell types, providing further evidence of their interactions in the TME of NPC.

Taken together, our intercellular interaction analysis revealed the complexity of cell-cell

communication network in NPC at a high dimension. Specifically, we identified the interactions between LAMP3+ DCs, Treg cells, and exhausted CD8+ T cells, which might jointly create an immune-suppressive niche for the TME of NPC (manuscript **Figure 7**; previous **Supplementary Figure 9**). Our findings provide insights into the development of therapeutic strategies for NPC by targeting the cell-cell interactions and thus interrupting the immune-suppressive niche.

In this revision, please kindly check the section “Intercellular interaction network in NPC” in the **Results**, the 2nd paragraph, Page 16 to the 1st paragraph, Page 17. We have added a brief introduction of CellPhoneDB in the **Methods** section. Please kindly check at the 2nd paragraph, Page 31.

8. Please identify 2 areas that have important meaning and then validate and check functionally for some of the markers as well.

Response: Thanks for the suggestion. First, our study identified multiple cell-cell interactions in the TME of NPC, including that of LAMP3+ DCs (DC_C3_LAMP3) and Treg cells as well as LAMP3+ DCs (DC_C3_LAMP3) and exhausted CD8+ cells (CD8_C11_PDCD1). Similar observations have also been reported in other type cancers, such as HCC (Zhang et al., *Cell* 2019, 179(4):829-845.e20) and PDAC (Jang et al., *Cell Rep* 2017, 20(3): 558-571). As suggested, we performed multiplex IHC assays, which revealed the co-localization of CD80-expressing LAMP3+ DCs (CD80+) and CTLA4-expressing Treg cells (CD3+CD4+FOXP3+), as well as PD-L1-expressing LAMP3+ DCs (CD80+) and PD1-expressing CD8+ T cells (CD3+CD8+), indicating direct contacts of the cells expressing these proteins (manuscript **Figure 6c** and **6d**, reproduced as **Response Figure 2A** and **2B**). Supportively, functional links between interacting pairs and corresponding cell types have been demonstrated previously in other cancers. DCs could restrict T-cell responses during the cross-presentation of tumour-antigens through PD-1/PD-L1 signalling (Sánchez-Paulete et al., *Ann Oncol* 2017, 28(suppl_12):xii44-xii55), and when the signalling was interrupted by PD-L1 deletion, tumour growth was significantly suppressed together with enhanced anti-tumour CD8+ T cell responses (Oh et al., *Nat Cancer* 2020, 1:681-691). It has also been reported that Treg cells could interact with DCs via their contacts between CTLA4 (on Treg cells) and the

costimulatory molecules CD80 and CD86 (on DCs) and promote the expression of IDO1 in DCs (Francesca et al., *Nat Immunol* 2003, 4(12):1206-12). IDO1 could block T cell responses to antigenic stimulation and induce the differentiation of Treg cells (Baban et al., *J Immunol* 2009, 183(4): 2475–2483). By contrast, the cross-talks between cytotoxic CD8+ T cells and either DCs or Treg cells have been widely addressed in multiple cancers. Taken together, we suspected that these cells jointly create an immune-suppressive niche for the TME of NPC.

Second, we identified both EBV positive (LMP1+EPCAM+) and EBV negative (LMP1-EPCAM+) malignant cells in NPC according to the transcription of EBV encoded molecules (LMP-1/BNLF2a/b, RPMS1/A73, LMP-2A/B and BNRF1; manuscript **Figure 5c**, reproduced below as **Response Figure 6A**). The existence of NPC cells with different EBV infection status was further validated using multiplex immunofluorescence staining assay with EPCAM as the marker of NPC cells arising from the epithelial cells (Meng et al., *Cell Death Dis* 2018, 9(1):2) and LMP1 (Epstein–Barr virus latent membrane protein 1) as the EBV infection marker (manuscript **Figure 5e**, reproduced below as **Response Figure 6B**). Given that all NPC tumours in the endemic regions are EBV positive, our findings are of particular interests for future studies on the role of EBV infection in the pathogenesis of NPC and the persistence of EBV in *in vivo* tumour cells compared to its gradual loss during *in vitro* culture.

We have added the related results into the manuscript in this revision. Please kindly check at the section “Intercellular interaction network in NPC” in the **Results**, the 2nd paragraph, Page 16 to the 1st paragraph, Page 17; and the section “Heterogeneity of malignant cells with different EBV infection status” in the 2nd paragraph, Page 14 to the 1st paragraph, Page 15.

Response Figure 6: Expression of EBV related genes and makers in malignant NPC cells.

A. UMAP plots showed the expression of EBV transcripts (LMP-1/BNLF2a/b, RPMS1/A73, LMP-2A/B, and BNRF1) in malignant NPC cells. Each dot represents a single cell. The depth of colour from grey to red represents low to high expression, respectively.

B. Representative images of immunofluorescence staining of malignant cells in NPC tissues. Proteins used in the assays are indicated on top. The red, green, orange arrows indicated the EPCAM, LMP1, and co-expression of EPCAM and LMP1 proteins in malignant cells, respectively. Scale bars, 50μm.

9. It would be helpful for the authors to try and assimilate the large body of data already existing instead of just new data that has no mechanistic support.

Response: Thanks for the suggestion. As suggested, we assimilated more existing data that are publicly available into our analyses, as additional supports or validations to our findings. These included single-cell RNA-Seq data for NPC (Zhao et al., *Cancer Lett* 2020, 477:131-143), NSCLC (Guo et al., *Nat Med* 2018, 24(7):978-985), HCC (Zhang et al., *Cell* 2019, 179(4):829-845.e20), and CRC (Zhang et al., *Cell* 2020, 181(2):442-459.e29), as well as bulk RNA-Seq data for an independent NPC cohort (n=113; Zhang et al., *Mol Cancer Res.* 2017, 15(12):1722-1732) and an in-house data of non-cancerous samples (rhinitis, n=10; Lin et al., *Lancet Oncol* 2020, 21(2):306-316).

First, we compared heterogeneous cell composition across different type of cancers and observed that infiltrating of multiple immune cells is common among cancers, while T cell is predominant in NPC (manuscript **Supplementary Figure 1g** and **1h**, reproduced as **Response Figure 4A** and **4B**). The heterogeneity of malignant cells is an important characteristic of NPC, which is reflected in Zhao et al.'s (Zhao et al., *Cancer Lett* 2020, 477:131-143) and our studies. In these two independent studies, malignant cells are both EPCAM positive and enriched in unique signalling pathways. NPC is a well-known EBV-associated malignancy (Chen et al., *Lancet* 2019, 394(10192):64-80), and we further divided malignant NPC cells into EBV+ or EBV- according to their expression of EBV transcripts. The presence of malignant NPC cells with different EBV infection status was further validated using multiplex immunofluorescence staining assay (manuscript **Figure 5e**, reproduced as **Response Figure 6B**).

Second, we observed similar expression profiles of signature genes in specific cell types across different cancers. For Treg cells, we identified a group of activated and immune-suppressive Treg cells in our NPC dataset, according to their functional status based on IL2R, inhibitory, and co-stimulatory scores (manuscript **Figure 3a**). Consistently, we also observed the activated and immune-suppressive Treg cells in CRC (Zhang et al., *Cell* 2020, 181(2):442-459.e29), HCC (Zhang et al., *Cell* 2019, 179(4):829-845.e20) and NSCLC (Guo et al., *Nat Med* 2018, 24(7):978-985; manuscript **Supplementary Figure 5c**, reproduced below as **Response Figure 7**). For LAMP3+ DCs, high expression of *CCR7* and *LAMP3*, which were related to their migration and maturation, were consistently observed in our NPC cohort, as well as in HCC and NSCLC (manuscript **Supplementary Figure 9a**, reproduced as **Response Figure 1B**). Moreover, we also found common expression of the immune-suppressive related genes (*CD274*, *PDCD1LG2*, *CD200*, *IDO1*, *EBI3*, *SOCS1*, *SOCS2*, and *SOCS3*) in LAMP3+ DCs among NPC, HCC, and NSCLC (manuscript **Supplementary Figure 9a**, reproduced as **Response Figure 1B**).

Third, we identified the ligand-receptor interaction of CTLA4-CD80 for Treg cells and LAMP3+ DCs, as well as CD274-PDCD1 (PD-L1-PD1) for LAMP3+ DCs and CD8+ T cells in NPC, using CellPhoneDB and multiplex IHC staining assays (manuscript **Figure 6c** and **6d**, reproduced as **Response Figure 2A** and **2B**). The former interaction for LAMP3+ DC and

exhausted CD8+ T cell was also demonstrated in HCC (Zhang et al., *Cell* 2019, 179(4):829-845.e20).

In summary, by assimilating more public data and additional verification assays, we further solidified our findings. In this revision, we have updated the abovementioned results. Please kindly check at the sections “The diversity and trajectory of Treg cells in NPC” in the 2st paragraph, Page 9, “Tumour-associated LAMP3+ DCs display a tolerogenic phenotype in NPC” in the 2nd paragraph, Page 13, and “Heterogeneity of malignant cells with different EBV infection status” in the 2nd paragraph, Page 14 to the 1st paragraph, Page 15.

Response Figure 7: Violin plots showed the functional status of CD4+ T cell cluster in CRC (colorectal cancer, A), NSCLC (non-small-cell lung cancer, B), and HCC (hepatocellular carcinoma, C).

The functional status was determined using the IL2R (top panel), inhibitory (middle panel), and co-stimulatory (bottom panel) scores of each CD4+ T cell cluster as indicated at the x-axis.

10. Placing such a large amount of data without any validation will just create more confusion in the field and the authors should make every effort to address this issue.

Response: Thanks for the comment and suggestion. By taking the suggestions from all the reviewers, we have performed additional analyses with public datasets and multiplex immunofluorescence staining to validate our findings in this revision. We note that our

findings were based on robust analytic pipelines that have been adopted in many studies, which reported positive results (Zhang et al., *Cell* 2019, 179(4):829-845.e20; Guo et al., *Nat Med* 2018, 24(7):978-985). We have also tried our best to present the findings with strong supporting evidence.

First, we presented our findings that could be generally observed in our NPC cohort, with any supporting evidence from other datasets or literature review. The cell composition of the tumour microenvironment (TME) was similar among NPC samples with inter-patient heterogeneity, which was consistent with the observations in other cancers (Lambrechts et al., *Nat Med* 2018, 24(8):1277-1289; Lee et al., *Nat Genet* 2020, 52(6):594-603; Peng et al., *Cell Res* 2019, 29(9):725-738).

Second, we performed multiple analyses to validate our findings. We integrated gene expression profiles, TCR tracing, chemokines expression, and sample origin to delineate the development and migration trajectories of T cells in NPC. We revealed that exhausted CD8+ T and immune-suppressive TNFRSF4+ Treg cells in NPC tumours were derived from CX3CR1+CD8+ T and naive Treg cells in peripheral blood, respectively (manuscript **Figure 2c** and **3c**). Consistently, Zhang et al. demonstrated the same trajectory of CX3CR1+CD8+ T cells, which migrated into the tumour from peripheral blood in NSCLC (Zhang et al., *Nature* 2018, 564(7735):268-272).

Third, we revisited our intercellular interaction data and revealed that the interactions were universal among the NPC cohort in the present study. The interaction between LAMP3+ DCs (DC_C3_LAMP3) and exhausted CD8+ T cells (CD8_C11_PDCD1) was observed in eight out of ten patients (manuscript **Supplementary Figure 11a**, reproduced as **Response 3A**), as well as the interaction between LAMP3+ DCs (DC_C3_LAMP3) and Treg_C4_TNFRSF4 in seven out of ten patients (manuscript **Supplementary Figure 11a**, reproduced as **Response 3A**). Moreover, the interactions were further consistently supported by their direct spatial contacts as revealed by the multiplex IHC staining assays (manuscript **Figure 6c** and **6d**, reproduced as **Response Figure 2A** and **2B**). Please kindly refer to our response to the Reviewer #1's point 3 above for details (Pages 4-8).

Besides, we found consistency between our findings and previous findings based on bulk RNA-seq. Please kindly refer to our response to the Reviewer #1's point 6 above (Pages

10-13).

In this revision, we have improved our manuscript, by incorporating the above information, and we have also provided more details for the relevant methods for better understanding.

Reviewer #2 (Remarks to the Author): Expert in scRNA-seq

This paper reports a comprehensive catalog of single-cell RNA expression phenotypes across 176,447 cells from 10 NPC tumors. The authors collate, correlate and filter the data with a variety of top of class algorithms to organize the data and derive various putative inferences. Among the many novel observations, the authors identify a subset of LAMP3+ dendritic cells that potentially exist in a (virtual) niche with Treg and exhausted CD8+ cells.

The study is comprehensive and unique. It is at this stage observational, and no experimental validation of any of the suppositions was performed. However, that should NOT delay publication of the manuscript as the effort will inform the work of many others. I would not recommend any follow up wet-lab experiments for this manuscript. The authors do not overstate their findings and keep the conclusions as postulates. I would however recommend that should the paper be accepted, the manuscript is carefully edited to keep the conclusions as "correlations" pending experimental validation.

Response: We are extremely grateful to the Reviewer #2 for the favourable comments on our study. As mentioned by the Reviewer #2, we applied analytic pipeline and algorithms in our study, of which robustness has been tested in many recent publications (Zhang et al., *Nature* 2018, 564(7735):268-272; Elyada et al., *Cancer Discov* 2019, 9(8):1102-1123; Vento-Tormo et al., *Nature* 2018, 563(7731):347-353). In this revision, we have performed additional analyses using some public datasets and managed to validate our findings. For example, our observations of LAMP3+ dendritic cells (manuscript **Supplementary Figure 9a**, reproduced as **Response Figure 1B**) and immune-suppressive Treg cells in NPC were consistent with that observed in other multiple cancers (manuscript **Supplementary Figure 5c**, reproduced as **Response Figure 7**). Moreover, we further validated cellular interactions between Treg and DC and CD8+ T cells, as well as malignant NPC cells with or without EBV

infection, using multiplex immunostaining assays (manuscript **Figure 5e** and **6c-6d**, reproduced as **Response Figure 2A-2B** and **6B**, respectively). Although these results once again suggest the robustness of our bioinformatics analyses, we acknowledge that our findings are pending further experimental validation.

We do appreciate the Reviewer #2's recommendation that experimental validation "should NOT delay publication of the manuscript as the effort will inform the work of many others". This was also our original intention. We wanted to publish our findings of various topics, so that these findings would therefore provide insights for the work of many scientists with different expertise. As such, scientists around the world would jointly discover exact molecular mechanisms regulating the development of NPC and figure out solutions for better control of NPC in the long run.

The figures are clear, the suppositions and conclusions they make are reasonable. Several of the algorithms are just listed as obscure names, like DEG. The paper would benefit when each new algorithm is introduced to give a brief explanation of the technique, and not just refer to the original paper. This will help others reading the paper to better follow the context of the conclusions.

Response: Thanks for the comments and suggestions. As suggested, we have revised the manuscript accordingly, with a brief explanation of each technique as well as the purposes of applications. Please kindly check at the **Methods** and **Results** sections.

I was curious, however, about why the authors did not search in the carcinoma cells (I presume they were in the dataset) for neo-antigens? Or is NPC not subject to neoantigen development? In either case, it should be noted.

Response: Thanks for the suggestion. Yes, there are 2,787 NPC cells with either positive or negative for EBV infection. We note that there is no study focusing on neoantigen prediction for NPC by far and literature searching against PubMed resulted in no relevant study (keywords: neoantigen nasopharyngeal carcinoma, as of Sep 4, 2020). Although previous studies have demonstrated a low mutation rate of NPC compared to other types of malignant tumours (Miao et al., *Mol Cancer* 2019, 18(1):128), high-frequency mutations of

genes (*NFKBIA*, *CYLD*, and *TP53*), splicing variants (*PTPRC*, *HIPK1*, and *TGFBR2*) and structural variations (*TNFAIP3*, *GEP192*, and *MED12L*) have been reported in NPC (Tu et al., *Carcinogenesis* 2018, 39(12):1517-1528; Li et al., *Nat Commun* 2017, 8:14121; Hong et al., *Proc Natl Acad Sci U S A* 2016, 113(40):11283-11288). It's plausible that these somatic alterations in cancer cells would result in the generation of novel proteins or neoantigens, therefore we believe that NPC is one of the malignancies that are shaped together with neoantigens and immunity. However, we are not able to predict any neoantigens in our present study, because we have no DNA level (whole genome or exome sequencing) data with these patient samples for HLA estimations and somatic mutations callings.

In this revision, we have acknowledged this in the main text. Please kindly check at the **Discussion** section, the 2nd paragraph, Page 21 to the 1st paragraph, Page 22.

Second, although the cohort size was modest, was there any clinical outcomes to which the expression profiles could be correlated?

Response: Thanks for the question. To address the Reviewer #2's comment, we updated clinical characteristics for all enrolled patients. Please kindly check the response to the Reviewer #3's minor point 1 below (manuscript **Supplementary Table 1**, reproduced as below **Response Table 1**, Pages 36-37). For the prognosis information, the average follow-up time was 17.375 months. Two patients died of NPC progression. With such a limited number of patients, we divided them into two groups, 'deceased' or 'alive'. Comparisons of cell content were tested between the two groups (**Response Figure 8A** and **8B**), which revealed the significantly increased proportion of CD8_C3_KLRB1 and decreased proportion of NK_C4_MKI67 in the tumours of the alive group (**Response Figure 8C**). However, because of the small cohort size in our study, we acknowledged that the above conclusion should be interpreted with caution and further confirmation is needed. Therefore, we ask to be allowed not to include these comparisons in our study.

Noteworthy, our study demonstrated that Treg_C4_TNFRSF4 had the strongest immune-suppressive function in NPC compared to the other two Treg cell clusters, with the highest expression levels of inhibitory molecules and co-stimulatory receptors. Moreover,

Treg_C4_TNFRSF4 also expressed *CCR8* specifically (manuscript **Supplementary Figure 5a**), which was known as a signature of immune-suppressive Treg cells that restrained immunity (Barsheshet et al., *Proc Natl Acad Sci U S A* 2017, 114(23):6086-6091). Therefore, we used the ratio of *CCR8* to *FOXP3* (the marker of Treg cells) as a surrogate for the proportion of Treg_C4_TNFRSF4 in Treg cells in NPC. We then performed survival analysis in another NPC cohort with 88 patients, which revealed that *CCR8/FOXP3* was significantly associated with the poor prognosis of NPC (manuscript **Supplementary Figure 5e**). Please kindly check at the section “The diversity and trajectory of Treg cells in NPC” in the **Results**, the 2nd paragraph, Page 10.

Response Figure 8: Association of cell composition and clinical characteristics of NPC.

A, B. Bar plots showed the difference in the proportion of each cluster from peripheral blood (A) or tumour (B) between alive and deceased groups. *P* value at the x-axis was tested by Two-sided Wilcoxon test.

C. Box plots showed the proportions of tumoral CD8_C3_KLRB1 and NK_C4_MKI67 in two groups of patients according to the prognosis status. Each comparison was tested by Two-sided Wilcoxon test.

I understand the authors don't want to call anything a "diagnostic" as that often invites criticism, but it would be good in the discussion to either mention whether there are any tentative conclusions worth following up in a larger cohort.

Response: Thanks for the understanding and suggestion. Indeed, our study revealed some novel findings that are potentially important for the clinical management of NPC.

First, the expression of *CCR8* and *FOXP3* might be promising diagnostic markers to stratify patients with NPC, since the survival analysis revealed that the ratio of *CCR8/FOXP3*, as a surrogate for the proportion of immune-suppressive Treg_C4_TNFRSF4 in Treg cells, was associated with poor survival of NPC (manuscript **Supplementary Figure 5e**). Please kindly refer to our response to the Reviewer #2's comment above (Pages 23-25).

Second, our findings also implicated potential immunotherapy for NPC as well as biomarkers. T cell exhaustion is one of the mechanisms of immune evasion adopted by cancer, often leading to inefficient control of cancers, and rejuvenation of exhausted T cells with inhibitory receptor blockade can promote immunity and favourable disease outcome. Intercellular interaction analysis revealed substantial immune-suppressive interactions between exhausted CD8+ T cells (*CD8_C11_PDCD1*) and LAMP3+ DCs (*DC_C3_LAMP3*), including *LGALS9-HAVCR2*, *CD200-CD200R*, *CD274-PDCD1*, and *PDCD1LG2-PDCD1* (manuscript **Figure 6b**; previous **Supplementary Figure 8C**). The immune-suppressive functions of *LGALS9* and *CD200* have been confirmed with the functional experiments in other cancers (Shintaro et al., *Recent Pat Endocr Metab Immune Drug Discov* 2013, 7(2):130-7; Jin et al., *Adv Exp Med Biol* 2020, 1223:155-165). Given that overall response rate of immunotherapies using the PD1 blockade is unsatisfactory (25.9-34%) according to the previous studies (Ma et al., *J Clin Oncol* 2018, 36(14):1412-1418; Hsu et al., *J Clin Oncol* 2017, 35(36):4050-4056), we propose that the application of *LGALS9* and *CD200* blockade, especially in combination with PD1 blockade, might be effective treatments to control NPC.

Third, our findings provide insights into the development of novel T cell therapy for NPC. Redirect the specificity of T cells has been shown to have significant anti-tumour activity (Sadelain et al., *Nature* 2017, 545(7655):423-431). We observed that *CX3CR1+CD8+* T cells in peripheral blood had highly cytotoxic potential and shared around 10% of the TCR clonotypes with the exhausted tumour infiltrating *CD8+* T cells (*CD8_C7-CD8_C11*;

manuscript **Supplementary Figure 4e**, previous **Supplementary Figure 3e**). This raises possible T cell therapies for NPC by infusing the CX3CR1+CD8+ T cells from peripheral blood after ex-vivo expansion or engaging T cells with CX3CR1 chemotactic potential towards the tumour site.

In this revision, we have updated the above information in the main text. Please kindly check at the 2nd paragraphs, Page 18 and the 1st paragraph, Page 21 in the **Discussion**. We also acknowledged that “we are not yet to draw a conclusion with such a limited cohort size that any components of the TME could be associated with the clinical outcomes”. Please kindly check at the 2nd paragraph, Page 21 to the 1st paragraph, Page 22 in the **Discussion** section.

This report will make a unique and important addition to the growing collection of human tumor/immune single cell datasets.

Response: Again, we appreciate the Reviewer #2 for the favourable comment on our study.

Reviewer #3 (Remarks to the Author): Expert in immuno-oncology

Major comments:

1. A recent paper on single cell profiling of tumor cells and infiltrating immune cells of NPC published in 2020 by Jin Zhao et al. in *Cancer Letters*, which decreases the novelty of the current work. So the statement of “first study” is not appropriate. I don’t think the authors cited and discussed the similar and novel findings compared to that study. For example, in the study Jin Zhao et al, NPC exhibited a high extent of intratumor heterogeneity, and malignant cell separated clearly in the tSNE map, which seems very different from the current data.

Response: Thanks for the comments. Previously, we overlooked the study by Zhao et al. (Zhao et al., *Cancer Lett* 2020, 477:131-143), which was published during our manuscript preparation. We apologized for this. In this revision, we have removed any statement of ‘first study’ of this kind and cited the study. Moreover, we have compared the findings between the two studies as suggested.

First, both studies revealed the intratumoral heterogeneity of NPC, containing majority of immune cells including T, B, myeloid, and NK cells as well as malignant epithelial cells. Our study further identified more than 53 subtypes among 176,447 cells from 10 patients with NPC, which is 56 times more than that reported by Zhao et al.'s study on three NPC tumour samples. This large-scale data enabled us to delineate the complexity of cell composition in NPC with much greater details.

Second, both studies reported common features of tumour infiltrating immune cells. For example, tumour infiltrating CD8+ T cells exhibited exhaustion markers (*PDCD1*, *CTLA4*, and *LAG3*). Diverse TCR repertoire and clonal expansion of TCR were also demonstrated. Moreover, both studies revealed intertumoral heterogeneity among patients. This was mainly represented by the different proportion of cell composition among patients (manuscript **Figure 1c**). For example, we observed that the proportion of tumour infiltrating T cells (CD4+ and CD8+) cells was different among patients, which is consistent with the observation in the Zhao et al. study.

Third, with the tumour-blood pair design and much large-scale data, we managed to profile the complexity of TME in NPC at a higher resolution, with novel findings. In brief, our study revealed the developmental trajectory of immune cells between peripheral blood and tumour site, with additional TCR repertoire sequencing, such as the differentiation of exhausted tumoral CD8+ T cells (CD8_C11_PDCD1) from CX3CR1+CD8+ T in peripheral blood (manuscript **Figure 2c** and **3c**). Moreover, we obtained potential cell-cell interaction networks among cell types, of which some were further validated using additional multiplex IHC staining assays. Please kindly refer to our response to the Reviewer #1's point 3 (Pages 4-8).

In summary, both Zhao et al.'s and our studies revealed the cell composition of TME of NPC, which showed intratumoral and interpatient heterogeneity. By contrast, our study provided a more detailed spectrum of cell composition of NPC tumour. Moreover, we delineated the developmental trajectories of infiltrating immune cells from peripheral blood. Furthermore, we identified intercellular interactions among different cell clusters. We have updated the comparisons accordingly in the revision. Please kindly check at the sections "Landscape view of cell composition in tumour biopsy and PBMC in patients with NPC", Page

5, "Heterogeneity of T cells and the diversity of TCR repertoire", Page 6 and the 1st paragraph, Page 7 in the **Results**.

Regarding the independent clustering of malignant cells for the three NPC samples in Zhao et al.'s study (**Figure 2A** in the original paper, which was reproduced as below **Response Figure 9A**), we suspected that such deviation might be attributed to the batch effect underlying their samples. We note that the authors did not describe the adjustment of the batch effect in their study. By communications with the authors, we were able to obtain and analyse the data using our pipeline. We performed canonical correlation analysis (CCA) and non-linear dimensional reduction of UMAP to remove the batch effect and reduce dimensionality before graph-based clustering, which revealed overlapped distribution of malignant cells for the three NPC samples (**Response Figure 9B**). On the other hand, we observed independent clustering of malignant cells from each NPC sample in our data when we applied the pipeline as stated in Zhao et al.'s study (**Response Figure 9C**). These observations suggest that it is necessary to detect and remove the confounding batch effect when handling associations between high-dimensional data sets (Ines et al., *BMC Syst Biol* 2016, 10(1):72.).

Response Figure 9. The distribution of malignant cells in Zhao et al.'s and our studies using different analytical methods.

A. t-SNE plot of the malignant cells in Zhao et al.'s study using PCA and t-SNE to reduce dimensionality. Each dot represents a cell, coloured according to the patient of origin. Please note that the figure was reproduced from the original paper for better reference only.

B. t-SNE plot of the malignant cells in Zhao et al.'s study using CCA and t-SNE to remove the batch effect and reduce dimensionality. Each dot represents a cell, coloured according to the patient of origin.

C. UMAP plot of the overall 2,787 malignant cells using PCA and UMAP to reduce dimensionality. Each dot represents a cell, coloured according to the patient of origin.

2. The authors spent most of efforts on the “intra-tumor heterogeneity”. What about the inter-patient heterogeneity? Did the TME components, NPC cells, interaction between

subsets of immune, between immune cells and NPC cells similar or different in each patient?
Is there any trend that age, gender, stage etc. impacts these relationships?

Response: Thanks for the inspirational questions. In this revision, we provided more comparisons among 10 patients with NPC, to reveal the interpatient heterogeneity of NPC.

Regarding the TME components of NPC, we observed identical composition of major cell types among all patients, with common infiltrating immune cells and predominance of T cells and B cells (**Response Figure 10A**). The proportions of each cell type are variable among individuals (**Response Figure 10A**). For example, the proportions range from 13.43% to 73.04 % for CD8+ T cell and 3.47% to 52.45% for B cells (manuscript **Figure 1c**).

Variable proportions of malignant NPC cell were also observed among different patients, ranging from 0.29 % to 20.60 % (**Response Figure 10A**). We further compared the gene expression profiles of malignant cells in each sample using GSVA, which revealed inter-patient heterogeneity of activated pathway. Cell cycle related pathways including E2F, MYC and G2M checkpoint were enriched in P02, P08, P11, P15, and P16 samples; TNF- α signalling via NF- κ B, angiogenesis, apoptosis, and EMT pathways were enriched in P06 sample; and Notch signalling pathway was enriched in P05, P06, P10, P13 and P15 samples (manuscript **Supplementary Figure 10f**, reproduced below as **Response Figure 10B**).

We next compared the interactions between subsets of immune cells and between immune cells and NPC cells among patients. We observed the interaction of LAMP3+ DC (DC_C3_LAMP3) and exhausted CD8+ T (CD8_C11_PDCD1) was commonly detected in eight out of ten patients via ligand-receptor pairs CD274-PDCD1 and PDCD1LG2-PDCD1 (manuscript **Supplementary Figure 11a**, reproduced below as **Response Figure 3A**). We also observed that the interactions (CX3CL1-CX3CR1, EGFR-TGFB1, and TNF-NOTCH1 and JAG1-NOTCH2) between EBV+ malignant cells (EP_C1_LMP1) and immune cells were detected in at least 50% of patients (manuscript **Supplementary Figure 12b-12d**, reproduced as **Response Figure 3B-3D**). However, the interacting intensity of the same ligand-receptor pair is variable among patients. These results suggest that the interactions between subsets of immune cells, between immune cells and malignant cells are common events but showing inter-patient heterogeneity in NPC.

We further examined whether the characteristics of patients would influence the results of inter-patient heterogeneity. We defined age groups as ‘high’ and ‘low’ according to the median age in these 10 patients. Clinical stage was determined according to the TNM classification for NPC (the 8th edition of the AJCC staging system; please kindly see **Methods**). We found that older patients had a significantly higher proportion of DC_C4_JCHAIN, NK_C3_TYROBP, and TH1_like_C1_CCR7 in peripheral blood and NK_C5_XCL1 in tumours (manuscript **Supplementary Figure 13a-13b**, reproduced as **Response Figure 11**); moreover, advanced patients at stage IV had a higher proportion of peripheral CD8_C10_MKI67 than patients at stages II/III (manuscript **Supplementary Figure 13c**, reproduced as **Response Figure 12**); B_C3_RGCC cells had a higher proportion of cell interactions in younger patients (manuscript **Supplementary Figure 13d**, reproduced as **Response Figure 13A and 13C**); DC_C4_JCHAIN and NK_C2_FCER1G had a significantly higher proportion of cell interactions in advanced patients at stage IV (manuscript **Supplementary Figure 13e**, reproduced as **Response Figure 13B and 13D**). We note that although there are statistical differences among groups, the tests were based on comparison groups with small sample sizes. In addition, all the patients we enrolled in this study were male and positive for EBV, which was excluded from the association test (manuscript **Supplementary Table 1**, reproduced as below **Response Table 1**)

In this revision, we added the description of inter-patient heterogeneity in the main text. We also acknowledge that “we are not yet to draw a conclusion with such a limited cohort size that any components of the TME could be associated with the clinical outcomes”. Please kindly check at the **Discussion**, the 2nd paragraph, Page 21 to the 1st paragraph, Page 22.

Response Figure 10: The inter-patient heterogeneity of NPC.

A. Bar plots showed the proportion of each cell type in tumour tissues of NPC.

B. Heatmap showed GSEA scores for the gene signatures associated with cancer-related functions in malignant cells from each patient as indicated at the bottom. The colour depth from blue to red indicates the GSEA score from low to high, respectively.

Response Figure 11: Association of age and the proportion of cell cluster.

A, B. Bar plots showed the difference in the proportion of each cluster from peripheral blood (A) or tumour (B) between age older and younger age groups. *P* value at the x-axis was tested by Two-sided Wilcoxon test.

C, D. The box plots showed the proportions of DC_C4_JCHAIN, NK_C3_TYROBP, and TH1_like_C1_CCR7 in peripheral blood (C) and NK_C5_XCL1 in tumour (D) were grouped according to the age among ten patients, which were tested by Two-sided Wilcoxon test.

Response Figure 12: Association of TNM stage and the proportion of cell cluster.

A, B. Bar plots showed the difference in the proportion of each cluster from peripheral blood (A) or tumour (B) between TNM II/III and IV groups. *P* value at the x-axis was tested by Two-sided Wilcoxon test.

C. Box plots showed the proportions of CD8_C10_MKI67 were grouped according to the TNM stage in peripheral blood among ten patients, which were tested by Two-sided Wilcoxon test.

Response Figure 13: Association of age or TNM stage and the proportion of interaction.

A, B. Bar plots showed the difference in the proportion of cell interaction in age (**A**) and TNM stage (**B**) groups. *P* value at the x-axis was tested by Two-sided Wilcoxon test.

C, D. Box plots showed the cell-cell interaction proportions of B_C3_RGCC, DC_C4_JCHAIN and NK_C2_FCER1G cells with other cells in all interactions of each patient with NPC grouped according to their age (**C**) or TNM stage (**D**), respectively.

Following are some minor comments and advices.

1. The patient information should be clearly presented since there are only 10 patients. Patient's characteristics may be important to interpret the data.

Response: Thanks for the suggestions. We have provided the patients' characteristics as a manuscript **Supplementary Table 1** in this revision, which was reproduced as below (**Response Table 1**).

Response Table 1. Clinical characteristics of 10 NPC patients in this study.

Num	Sample ID	Age	Gender	Pathological diagnosis*	EBERS hybridization	TNM stage*	Metastasis	Collection date	Last follow-up date	Follow-up month	Prognosis of status#
1	P02			Undifferentiated non-keratinized carcinoma	+	T3N2M0, III	Cervical lymph node	2018-6	2019-7	13	0
2	P05			Undifferentiated non-keratinized carcinoma	+	T4N3M1, IVb	Cervical lymph node, shoulder	2018-6	2020-3	21	1
3	P06			Undifferentiated non-keratinized carcinoma	+	T4N3M0, IVa	Cervical lymph node	2018-6	2020-3	21	1
4	P07			Undifferentiated non-keratinized carcinoma	+	T3N2M0, III	Cervical lymph node	2018-7	2020-3	20	1
5	P08			Undifferentiated non-keratinized carcinoma	+	T4N3M0, IVa	Cervical lymph node	2018-7	2020-5	22	1
6	P10		[redacted]	Undifferentiated non-keratinized carcinoma	+	T3N1M0, III	Cervical lymph node	2018-8	2020-3	19	1
7	P11			Undifferentiated non-keratinized carcinoma	+	T3N2M0, III	Cervical lymph node	2018-8	2020-3	19	1
8	P13			Differentiated non-keratinized carcinoma	+	T2N2M0, III	Cervical lymph node	2018-9	Lost to follow-up	NA	NA
9	P15			Undifferentiated non-keratinized carcinoma	+	T2N1M0, II	Cervical lymph node	2018-9	Lost to follow-up	NA	NA
10	P16			Undifferentiated non-keratinized carcinoma	+	T3N3M1, IVb	Cervical lymph node, parotid lymph node	2018-9	2019-1	4	0

*:Pathological diagnosis and TNM stage of NPC were determined according to the 8th edition of the International Union against Cancer (UICC) and American Joint Committee on Cancer (AJCC) staging system.

#:0 stands for deceased, and 1 stands for alive.

2. Although NPC is was EBV tested in these patients either by antibody or PCR

Response: Thanks. Yes, we had tested EBV infection status in tumour tissues of these patients using in situ hybridization for EBV encoded small RNAs (EBERs). All 10 tumour samples were EBV-positive. A representative image was provided as manuscript **Supplementary Figure 1a** and reproduced as below **Response Figure 14**.

Response Figure 14: Representative image of the *in-situ* hybridization of EBERs.

Brown staining indicates scattered EBER-positive malignant cells in a case of NPC. Scale bars, 100 μ m.

3. Most patients had stage III-IV disease. Was the tissues from FNA or core biopsy?

Response: Thanks for the question. Tumour tissues were obtained using endoscopic nasopharyngeal biopsy. The information was added in the revision. Please kindly check at the section “Patient recruitment and sample collection” in the **Methods**, the 1st paragraph, Page 23.

4. How to distinguish doublet and single cells in supplementary figure 1B?

Response: Thanks for the question. We used an R package “DoubletFinder” to predict doublets in single-cell RNA sequencing data, which was artefactual libraries generated from two cells arising due to errors in droplet encapsulation of cell. Briefly, a doublet is defined as a single-cell library representing more than one cell, and a closer examination of some known markers would suggest that the offending cluster consists of doublets of more than

one cell type, while no cell type is known to strongly express both of these markers at the same time. DoubletFinder predicts doublets according to each real cell's proximity in gene expression space to artificial doublets created by averaging the transcriptional profile of randomly chosen cell pairs. The remaining cells survived from the filtering criteria are single cells.

In this revision, we provided more details descriptions regarding the methodology in the **Methods** section and a brief description in the figure legend of manuscript **Supplementary Figure 1c** (previous **Supplementary Figure 1B**). Please kindly check at the section "Single-cell gene expression quantification and determination of cell types", the 2nd paragraph, Page 25.

5. The study identified 176,447 cells. All cells types in TME and blood should be included, however, there were no fibroblasts and endothelial cells that are commonly found in TME of other cancer types. What was the reason?

Response: Thanks for the comment. We indeed detected a few fibroblasts and endothelial cells in our database, but they were not identified as clear clusters. There are 56 fibroblasts with typical makers *COL1A1* and *COL3A1*, and seven endothelial cells with markers *FLT1* and *VWF* (manuscript **Supplementary Figure 1f**, reproduced below as **Response Figure 15**). Such small number of fibroblasts and endothelial cells in our dataset might be due to that 1) the TME of NPC might contain a low content of fibroblasts and endothelial cells and/or 2) the single-cell preparation and capture process might have a certain preference on cell types, resulting in low representation of fibroblasts and endothelial cells. In addition, we note that neither fibroblasts nor endothelial cells were identified by Zhao et al.'s study, which reported single-cell RNA sequencing on three NPC tumours (Zhao et al., *Cancer Lett* 2020, 477:131-143).

In this revision, we added that "in addition, we detected 56 fibroblasts and seven endothelial cells with sparse distribution among TME cells (manuscript **Supplementary Figure 1f**), which might reflect their intrinsic nature in NPC or their low representation due to the technical limitations." Please kindly check at the section "Landscape view of cell

composition in tumour biopsy and PBMC in patients with NPC” in the **Results**, the 1st paragraph, Page 5.

Response Figure 15: UMAP plots showed the normalized expression of markers for fibroblasts (*COL1A1* and *COL3A1*) and endothelial cells (*FLT1* and *VWF*).

Each dot represents a single cell and the depth of colour from grey to blue represents low to high expression, respectively.

6. Line 110, what are the ‘epithelial cells’? Malignant cells? Non-malignant cells?

Response: Thanks. Previously, the ‘epithelial cells’ included EP_C1_LMP with high expression of EBV molecules and EP_C2_EPCAM with high expression of *EPCAM*, which were considered as malignant cells and non-malignant cells, respectively. In this revision, we performed additional analysis using inferCNV as suggested by the Reviewer#3 (please kindly refer to below **minor point 9**, Pages 41-44) and observed that all these cells have aberrant copy number alterations, which suggest that they are malignant cells. Therefore, we have updated the new results in our manuscript accordingly. The ‘epithelial cells’ are malignant cells with either high or low expression of EBV molecules and aberrant CNVs. Please kindly

check at the section “Heterogeneity of malignant cells with different EBV infection status” in the **Results**, the 2nd paragraph, Page 14 to the 1st paragraph, Page 15.

7. What are the criteria to select the ‘gene’ at the end of different T/B cell cluster names.

Response: Thanks. The selection criteria for one ‘gene’ as cluster name includes 1) with top ranking at the differential gene expression analysis for the corresponding cell cluster, 2) with strong specificity of gene expression meaning high expression ratio within the corresponding cell cluster but low in other clusters, and 3) with literature supports that it’s either a marker gene or functional relevant to the type of cell. In this revision, we have updated the nomenclature of cell cluster in the **Methods** section. Please kindly check at the section “Single-cell gene expression quantification and determination of cell types”, the 3rd paragraph, Page 26 to the 1st paragraph, Page 27.

8. Line 221, please clarify how to calculate the IL2R scores in ‘Method’ section.

Response: Thanks for the suggestion. We used the ‘AddModuleScore’ function in Seurat software to calculate the IL2R scores, which considered the average expression levels of IL2RA, IL2RB, and IL2RG in each cell of a cluster, subtracted by the aggregated expression of control feature sets. All genes were binned based on their average expression, and the control features were randomly selected from each bin. As suggested, we have added the description in the **Methods** section in this revision. Please kindly check at the section “Calculation of functional module scores”, the 3rd paragraph, Page 27.

9. It is not convincing to define EP_C1_LMP1 as NPC tumor cells, provide more approaches such as inferred CNVs.

Response: Thanks for the suggestion. As suggested, we used inferCNV to identify evidence for somatic large-scale chromosomal copy number alteration in epithelial cells (manuscript **Figure 5a**, reproduced below as **Response Figure 16A**). With reference to single-cell data of normal nasopharyngeal epithelium (GEO accession number: GSE121600; Ruiz et al., *Development* 2019, 146(20):dev177428), both clusters of EBV+ (previous EP_C1_LMP1) and EBV- (previous EP_C2_EPCAM) epithelial cells exhibited higher CNV scores than normal

nasopharyngeal epithelium and were thus identified as malignant cells. In addition, we performed inferCNV with tumour infiltrating CD8+ T cells and CD8+ T cells in peripheral blood (**Response Figure 16B**) and did not observe any large-scale chromosomal CNV, suggesting the robustness of the inferCNV analysis.

In this revision, we have updated the corresponding information in **Methods** and **Results** sections. Please kindly check at the sections “inferCNV analysis”, the 3rd paragraph, Page 30 to the 1st paragraph, Page 31, and “Heterogeneity of malignant cells with different EBV infection status”, the 2nd paragraph, Page 14 to the 1st paragraph, Page 15.

Response Figure 16: InferCNV analysis of malignant cells and CD8+ T cells.

A. Heatmap showed large-scale CNVs for epithelial cells (rows) from ten NPC tumours, inferred based on the average expression of 100 genes surrounding each chromosomal position (columns). Red: gains; blue: losses. The references derived from normal nasopharyngeal epithelium (top panel) and NPC cells (lower panel) are coloured according to each patient.

B. Heatmap showed large-scale CNVs for CD8+ T cells (rows) from ten tumours, inferred based on the average expression of 100 genes surrounding each chromosomal position (columns). Red: gains; blue: losses. The references are CD8+ T cells derived from peripheral blood samples (top panel) matching to the patients with NPC (lower panel).

10. Suggest plotting the epithelial cells by patients in addition to Figure 5A.

Response: Thanks for the suggestion. We have updated the corresponding figure (manuscript **Supplementary Figure 10a**) in this revision.

11. Describe intertumoral heterogeneity and intratumoral heterogeneity of malignant cells of NPC.

Response: Thanks for the suggestion. We identified a total of 2,787 NPC malignant cells and divided them into EBV+ and EBV- malignant cell groups according to their expression of EBV-encoded transcripts (manuscript **Figure 5b** and **5c**, reproduced as **Response Figure 5A** and **6A**). We observed higher expression of chemokines including *CXCL3*, *CXCL8*, and *CX3CL1* in EBV+ malignant cells compared to EBV- malignant cells (manuscript **Figure 5d**, reproduced as **Response Figure 5B**). Moreover, intercellular interaction analysis revealed that EBV+ malignant cells had more interactions with immune cells compared to EBV- malignant cells (manuscript **Supplementary Figure 12a**, reproduced as **Response Figure 5C**). To better characterize the intratumoral heterogeneity of malignant NPC cells, we further performed clustering analysis of all malignant cells, which revealed five prominent cell subgroups (C1-C5; manuscript **Supplementary Figure 10d**, reproduced below as **Response Figure 17A**). We deciphered the molecular characters of malignant cells in different clusters, using gene set variation analysis (GSVA), which revealed distinct enrichment of signalling pathways in each subgroup. Specifically, C1 cluster was enriched with hypoxia, apoptosis, P53, TGF-beta, and WNT signalling pathways; C2 and C5 clusters were enriched with EMT and angiogenesis pathways; C3 cluster was enriched with oxygen species pathway; and C4 cluster was enriched with MYC, E2F, and G2M checkpoint pathways (manuscript **Supplementary Figure 10f**, reproduced below as **Response Figure 17B**).

The intertumoral heterogeneity of malignant NPC cells is reflected by the different proportions of cell subtypes. The proportions of EBV+ in all malignant NPC cells are variable among patients, ranging from 35.00% to 87.20 % (manuscript **Supplementary Figure 10e**, reproduced below as **Response Figure 17B**). Moreover, the P02, P08, P11, and P15 samples have a higher proportion of high proliferative cluster (C4 cluster) than other samples (manuscript **Supplementary Figure 10e** and **10g**, reproduced below as **Response Figure 17C** and **17D**). Likely because of the consisting of different proportions of subtypes in malignant cells, each tumour showed distinct enrichment of signalling pathways according to gene set variation analysis. For example, the above four samples having a higher proportion of C4 cluster showed enrichment of cell cycle related pathways including E2F, MYC and G2M checkpoint (manuscript **Supplementary Figure 10e** and **10f**, reproduced as **Response Figure 10B**). Please kindly refer to our response to the Reviewer #3's major point 2 above (Pages 30-36).

In this revision, we have updated the above information in the **Results** section. Please kindly check at the section "Heterogeneity of malignant cells with different EBV infection status", the 2nd paragraph, Page 15 to the 1st paragraph, Page 16.

Response Figure 17: The heterogeneity of malignant NPC.

A. UMAP plot of 2,787 malignant NPC cells grouped into five cell subtypes according to refined clustering analysis. Each dot represents a cell, coloured according to cell types.

B. Heatmap showed GSVAscores for the gene signatures relevant to carcinogenesis in malignant cells from each cluster as indicated at the bottom. The colour depth from blue to red indicates GSVAscore from low to high, respectively.

C. Bar plots showed the proportion of EBV+ and EBV- malignant cells (top panel) as well as in-depth cell sub-cluster by clustering analysis in NPC (bottom panel).

D. Violin plot showed the proliferation scores of each malignant cell cluster in **A.** calculated by AddModuleScore function in Seurat. Cell clusters are indicated at the x-axis, and proliferation score is indicated at the y-axis.

12. Line 567-568, please mind the unidentified characteristics.

Response: Thanks for pointing out the unidentified characteristics and the suggestion. The content should be “enzymatic type II (Thermo Fisher Cat. no. 17101015) and IV (Thermo Fisher Cat. no. 17104019)”, and the Roman numbers II and IV were not identified by the manuscript conversation system. In this revision, we have corrected this issue and make sure all characters are well presented.

Reviewer #4 (Remarks to the Author): Expert in immuno-oncology

The authors present single cell RNA data for nasopharyngeal carcinoma for tumor and peripheral blood. While this data is an important road map to understand cell signatures with this type of cancer, the data are highly descriptive with too many suggestive correlations that are not backed up with sound science. In particular, there is only one sub-figure that shows protein expression to back up the RNA data. With the paucity of validating protein data it is hard to interpret most of the claims in this manuscript. Hence, most of what is claimed is suggestive at best.

Response: Thanks for the comments. In the current study, we generated comprehensive profile of the TME of NPC using single-cell transcriptome analysis of more than 53 cell subtypes widespread in NPC samples, which provides an important road map to understand the cell signatures of NPC, as pointed out by the Reviewer #4. Moreover, we delineated the developmental links and cell-cell interactions among peripheral, tumoral infiltrating immune cells and heterogeneous malignant cells in NPC with the high resolution. These would provide clues for further studies by scientists with different expertise to understand the pathogenesis of NPC and develop therapeutic strategies for NPC. It is indeed our primary intention since we don't have all the expertise of different disciplines at this moment. Supportively, the Reviewer #2 addressed that **“that (no experimental validation) should NOT delay publication of the manuscript as the effort will inform the work of many others. I would not recommend any follow up wet-lab experiments for this manuscript.”** Please kindly refer to the Reviewer #2's comment above (Pages 21-27). Furthermore, we assured that our findings are robust, which were obtained using large-scale data of tumour-blood

pairs design, sophisticated and proven analytic pipeline, and cross-validations with multiple analyses, multiple samples, and literature review. Studies on other cancers with a similar design have been published in many prestigious journals (Zhang et al., *Cell* 2019, 179(4):829-845.e20; Guo et al., *Nat Med* 2018, 24(7):978-985; Zhang et al., *Nature* 2018, 564(7735):268-272). In this revision, we further provided additional validations using public data and multiplex immunostaining assays.

First, our study revealed the development trajectory of tumoral infiltrating T cells from peripheral blood in NPC, using multiple analyses incorporated with transcriptome, TCR sequence tracing, and sample origin. With these analyses, we discovered that exhausted CD8⁺ T and immune-suppressive TNFRSF4⁺ Treg cells in NPC tumours were derived from CX3CR1⁺CD8⁺ T and naive Treg cells in peripheral blood, respectively (manuscript **Figure 2c** and **3c**). Consistently, the migration of peripheral CX3CR1⁺CD8⁺ T cells into tumour followed by differentiation into cytotoxic CD8⁺ T cells and exhausted CD8⁺ T cells has been well documented in other cancers (Guo et al., *Nat Med* 2018, 24(7):978-985; Zhang et al., *Nature* 2018, 564(7735):268-272).

Second, we identified extensive cell-cell interactions in the TME of NPC using CellPhoneDB software and further validated profound immune checkpoint interactions (CTLA4-CD80 and PD-L1-PD1) among LAMP3⁺ DCs (DC_C3_LAMP3), Treg cells, and exhausted CD8⁺ T cells (CD8_C11_PDCD1), using multiplex immunohistochemistry staining (manuscript **Figure 6c** and **6d**, reproduced as **Response Figure 2A** and **2B**). Supportively, similar PD-L1-PD1 interaction between LAMP3⁺ DCs and CD8⁺ T cells has been reported in HCC (Zhang et al., *Cell* 2019, 829-845.e20). The interaction between CTLA4-CD80 could promote the expression of *IDO1* in DCs (Francesca et al., *Nat Immunol* 2003, 4(12):1206-12). *IDO1* could block T cell responses to antigenic stimulation and induce the differentiation of Treg cells (Baban et al., *J Immunol* 2009, 183(4): 2475–2483). Taken together, we suspected that LAMP3⁺DCs, Treg cells and exhausted CD8⁺ T cells jointly create an immune-suppressive niche for the TME of NPC.

Third, we identified and validated the existence of NPC cells with different EBV infection status using single-cell expression profile and multiplex immunofluorescence staining assays. EBV⁺ malignant cells (EP_C1_LMP1) exhibited higher activation of NF- κ B and Notch

pathways and higher expression of multiple chemokines (*CX3CL1*, *CXCL3* and *CXCL8*) than EBV- malignant cells (EP_C2_EPCAM; manuscript **Figure 5d**, reproduced as **Response Figure 5B**). Consistently, altered expression and accumulated mutations in NF- κ B and Notch pathways have been commonly implicated in EBV infection and tumorigenesis in NPC (Li et al., *Nat Commun* 2017, 8:14121; Tu et al., *Carcinogenesis* 2018, 39(12):1517-1528; Shi et al., *Int J Cancer* 2006, 119(10):2467-75). These findings again indicate the significant contribution of EBV infection and its related NF- κ B and Notch pathways to NPC. We note that EBV- malignant NPC cells make up the heterogeneity of NPC. Further studies would be necessary to address the development of the cells and their contribution to variable treatment responses.

Lastly, our results were generated based on reliable and rigorous analyses. In our present study, we applied robust algorithms ('top of class' as commented by the Reviewer #2) in the analytic pipelines, which have been widely adopted in many studies (Zhang et al., *Cell* 2019, 179(4):829-845.e20; Guo et al., *Nat Med* 2018, 24(7):978-985; Zhang et al., *Nature* 2018, 564(7735):268-272; Elyada et al., *Cancer Discov* 2019, 9(8):1102-1123). Moreover, we performed multiple analyses using the expression profiles, ligand-receptor relationship, TF regulatory information and TCR tracing to explore the TME of NPC. As such, the results based on the cross-validations should be very likely robust. For example, we were able to confirm the spatial contacts of DC, Treg and CD8+ T cells in the TME of NPC. Furthermore, our findings were reproducible in individual patients in our cohort (manuscript **Supplementary Figure 11a** and **12b-d**, reproduced as **Response Figure 3**). Besides, we assimilated other public data and observed similar cell types and cell signatures among NPC, colorectal, liver, and lung cancers. Please kindly refer to our response to the Reviewer #1's point 9 (Pages 17-19).

In summary, we presented a comprehensive single-cell transcriptome analysis of NPC, which revealed notable cell-cell interactions among LAMP3+ DCs, Treg cells, and exhausted CD8+ T cells and the heterogeneous malignant cells with different EBV status in the TME of NPC. The robustness of our findings was further confirmed using additional experiments (multiplex immunofluorescence or immunohistochemistry staining assays) and analyses with public data in this revision.

There are also techniques/signatures that are not explained very well so it was extremely hard to review some of the data as it was not clear what the authors' were claiming. In summary, if the authors resubmit this manuscript there either has to be protein data to back up their claims or the manuscript needs to be rewritten to take out all of the unsubstantiated claims throughout the results and discussion sections.

Response: Thanks. We apologised that we omitted the introduction of some techniques and signatures in our previous submission. As suggested, we provided introductions in more details for each key technique or signature, as well as the purposes of the applications. Moreover, we included additional validation results derived from multiplex immunostaining assays and analyses with the assimilation of public datasets. We believe that the presentations of our findings have been improved and could be clearly followed in this revision. We have also softened our claims of findings and acknowledged that further functional experiments are pending to validate and explore the underlying mechanisms. Please kindly check at the **Methods, Results** and **Discussion** sections.

REVIEWERS' COMMENTS

Reviewer #3 (Remarks to the Author):

The authors have addressed my questions.
Thanks for the efforts.

Reviewer #5 (Remarks to the Author):

It is not the first single cell transcriptome landscape study in NPC, but it is the most comprehensive analysis. I can see that the authors spent great deal of time characterizing LAMP3+ DC populations. Please cite the single cell RNA sequencing study on NPC by Chen et al. (Cell Res. 2020 Nov;30(11):1024-1042.

Point-by-point responses to reviewers' comments

Reviewer #3 (Remarks to the Author):

The authors have addressed my questions.
Thanks for the efforts.

Thank you.

Reviewer #5 (Remarks to the Author):

It is not the first single cell transcriptome landscape study in NPC, but it is the most comprehensive analysis. I can see that the authors spent great deal of time characterizing LAMP3+ DC populations. Please cite the single cell RNA sequencing study on NPC by Chen et al. (Cell Res. 2020 Nov;30(11):1024-1042.

Thanks for your favorable comments. We have cited the reference in “Introduction” section (reference 18).